# Learning from imagined experiences via an endogenous prediction error

Aroma Dabas [1,2] ✉, Rasmus Bruckner [3,4], Heidrun Schultz [1,5],
Frederik Bergmann [1,6] & Roland G. Benoit [1,6] ✉

Experiences shape preferences. This is particularly the case when they deviate from our expectations and thus elicit prediction errors. Here we show that prediction errors do not only occur in response to actual events – they also arise endogenously in response to merely imagined events. Specifically, people repeatedly chose between different acquaintances and then imagined interacting with them. Our results show that they acquired a preference for acquaintances with whom they had pictured unexpectedly pleasant events. This learning can best be accounted for by a computational model that calculates prediction errors based on these rewarding experiences. Using functional MRI, we show that the prediction error is mediated via striatal activity. This activity, in turn, seems to update preferences about the individuals by updating their cortical representations. Our findings demonstrate that imaginings can violate our own expectations and thus drive endogenous learning by coopting a neural system that implements reinforcement learning.

A hallmark of the human mind is its ability to adapt to an ever-changing environment[1]. Our preferences are largely not hard-wired but continuously shaped by experiences. These update the values we assign to objects as well as to the places and people that we encounter in our environment[2]. Much of such learning is driven by a mismatch between our expectations and actual experiences[3]. For example, when we have an unexpectedly rewarding experience with a particular person, this enhances our expectation that a repeated encounter will also be positive.

Models of reinforcement learning (RL) have formalized this mismatch as a prediction error (PE)[3,4]. This PE serves as a teaching signal that elicits the updating of our values and preferences. We here hypothesize that RL does not only occur in response to external events that we experience[5,6] or witness[7,8]. Instead, we suggest that it also arises endogenously as a consequence of internal events that we have merely imagined.

The capacity to imagine hypothetical events is often referred to as episodic simulation. It shares several features with episodic memory: The two capacities exhibit parallel developmental trajectories[9,10], are similarly affected by lesions to the medial temporal lobes[11,12], and are also more broadly supported by the same core network of brain regions[13]. These commonalities have been taken to suggest that episodic simulation is grounded in our memory systems[14,15]. These provide the building blocks for our simulations as well as the constructive processes to recombine these building blocks into novel events.

Given the functional overlap between episodic simulation and memory, we hypothesize that we can also learn from simulated experiences, much like we learn from actual experiences. Indeed, motor imagery can enhance motor performance[16] and aversive mental imagery elicits de novo fear conditioning[17]. Moreover, when we simply imagine chance encounters with beloved people at random locations, this changes how much we like the location of the imaginary meetings[18,19].

Such simulation-based learning does not entail any actual feedback from the environment—yet we hypothesize that it induces learning much in the same way as experience-based learning. Episodic simulations can lead to the consideration of alternative outcomes[20,21] and forge new insights[22,23]. They can thus yield mismatches with our prior beliefs. On an algorithmic level, we hypothesize that such

[1]Max Planck Institute for Human Cognitive and Brain Sciences, Leipzig, Germany. [2]Humboldt University of Berlin, Berlin School of Mind and Brain, Berlin, Germany. [3]Freie Universität Berlin, Berlin, Germany. [4]University of Hamburg, Hamburg, Germany. [5]Dresden University of Technology, Dresden, Germany. [6]University of Colorado Boulder, Boulder, CO, USA. ✉e-mail: dabas@cbs.mpg.de; roland.benoit@colorado.edu

internally generated mismatches cause PE, inducing learning in a manner similar to that of externally derived mismatches.

On a neural level, we also hypothesize that this simulation-based RL is based on a mechanism that is shared with experience-based RL. Specifically, the endogenous PE should then also be mediated via dopaminergic activity in the ventral striatum, as demonstrated in neurophysiological recordings in rodents[24,25] and monkeys[26,27] and consistent with regional activation observed with human fMRI[28,29].

We further suggest that this striatal activity enables learning about an environmental stimulus by inducing plasticity in its cortical representation[30,31]. For example, the dorsomedial prefrontal cortex (dmPFC) encodes representations of familiar people[32–34]. We would thus expect the ventral striatum to interact with this region when we experience an unexpected reward while imagining someone that we know.

To test these complementary hypotheses, we developed a procedure that is akin to a stable two-armed bandit task (Fig. 1). In preparation for this task, participants first provided a list of people with whom they are personally familiar. They then rated how much they liked each person. Four of the most neutrally liked people were allocated to a high-reward (HR) and a low-reward (LR) condition.

During the fMRI session, participants repeatedly made choices between two people, one from each reward condition (e.g., Sally vs. Harry). They then imagined a vivid interaction with the chosen person (e.g., Sally) in a specified scenario. These scenarios were either pleasant (e.g., "Sally wishes you well on your birthday") or neutral to unpleasant (e.g., "Sally returns your bike broken"). The people in the HR condition had a higher probability to be presented with pleasant scenarios than

those in the LR condition (80% vs. 30% of the trials). After each trial, participants indicated the pleasantness of the imagined interaction. We take this outcome measure as a proxy for the experienced reward and use it to model the PE on a given trial.

This procedure allowed us to test key features of our hypotheses. First, on a behavioral level, we show that participants acquire a preference for people in the HR, over people in the LR, condition. This preference shift seems to reflect a value update of the people, as it also correlates with an external measure of how much they were liked after learning, relative to before. Second, on an algorithmic level, the preference shift is best accounted for by the Rescorla-Wagner (RW) model[4] of RL. Third, on a neural level, this endogenous RL is also implemented in a similar fashion as experience-based RL. Specifically, a model-based analysis of our fMRI data indicates that the endogenous PE is mediated by neural activity in the ventral striatum. Moreover, using representational similarity and psychophysiological interaction analyses, we provide evidence that the striatal PE updates value by interacting with cortical areas that are involved in representing the information that is being updated. In the case of information about familiar people, this is particularly the case for the dmPFC[32–34].

## Results
### Episodic simulations shift preferences
We first assessed whether participants acquired a preference for selecting people with whom they had mentally experienced more pleasant episodes. To test this prediction, we calculated, for each participant, the overall probability of choosing the HR people across the experimental session. These probabilities were larger than chance,

### a  List of familiar people

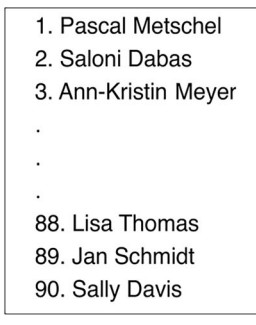

### b  Pre- & post ratings

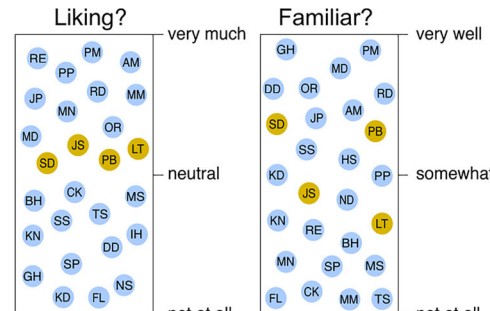

### c  Episodic simulation (MRI)

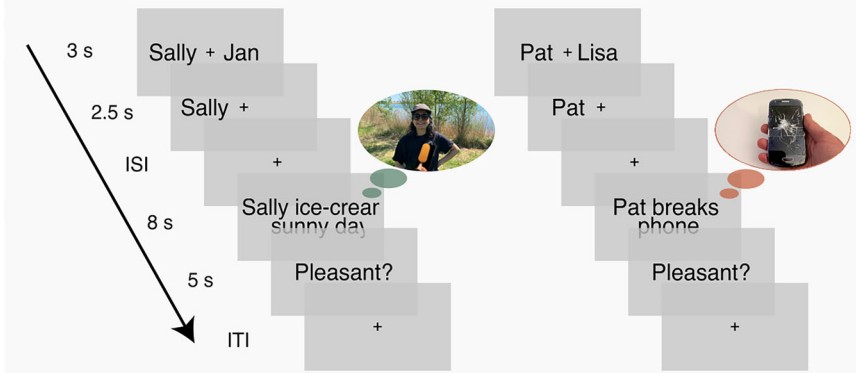

**Fig. 1 | Experimental procedure. a** Participants provided names of personally familiar people, and **b** rated them according to their liking and familiarity—both before and after the episodic simulation task. Based on the initial rating, we selected the six most neutrally liked people, two each to create a high reward (HR), a low reward (LR), and a baseline condition. **c** On each trial in the MRI scanner, participants made a choice between an HR and an LR person. They sought to select the person that is likely to yield a more pleasant imagined episode. After making their decision, they imagined interacting with the person in either a pleasant (probability ratio: HR/LR = 0.8/0.3) or neutral-to-unpleasant scenario. Participants then rated the pleasantness of the simulated interaction, which served as a proxy for the reward value of the mental experience.

## a  Simulations induce a shift in preferences

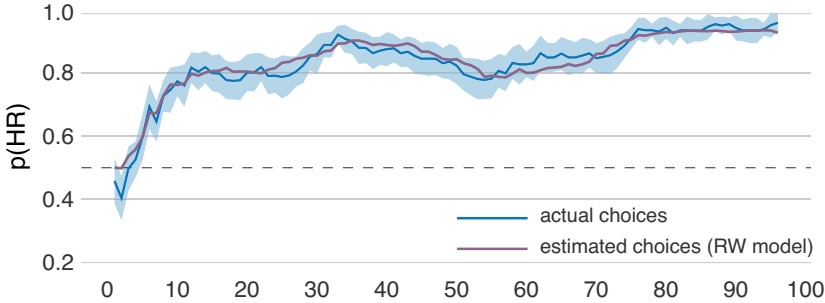

## b  Greater probability of choosing high-reward people

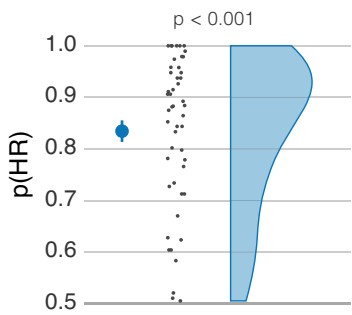

## c  Learning also manifests as value update

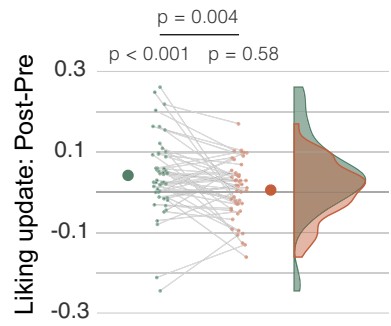

## d  Preference shift correlates with value update

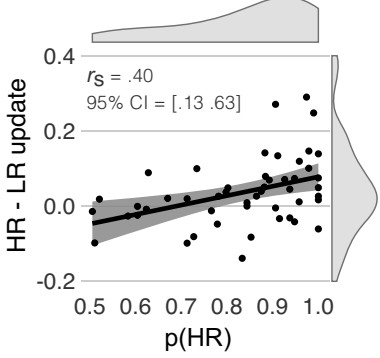

## e  Rescorla-Wagner model accounts best for learning

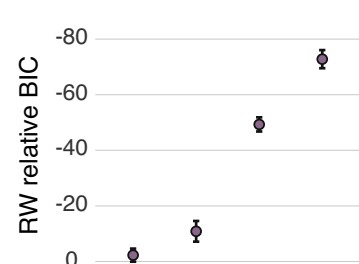

**Fig. 2 | Episodic simulation induces learning. a** Over the course of the experiment, participants acquired a preference for selecting people from the high-reward (HR) condition (blue line). The shaded blue area indicates the standard error of mean. The trial-wise estimated choices from the Rescorla-Wagner (RW) model showed a good fit with the empirical data (purple line). **b** Overall, participants showed a high probability for selecting an HR (versus a low-reward; LR) person. Statistical significance was assessed using a one-tailed Wilcoxon signed-rank test; $p = 5.6e\text{-}10$. **c** On the external rating task, episodic simulations also led to an increase in liking for persons in the HR condition that was greater than the absent change in liking for the LR condition. **d** The acquired preference for HR

people in the simulation task (p(HR)) correlated with the increase in liking for HR (vs. LR) people across participants, suggesting that both effects reflect a value update induced by the episodic simulations. Black line shows the linear regression and the shaded area the 95% confidence interval. **e** Relative to the RW model, all other models had a poorer fit to the empirical data. Model fit was estimated using BIC values, and the difference was computed as $BIC_{RW} - BIC_{model}$. Larger dots denote the means; the error bars denote the standard error of the mean; $n = 49$; CK Choice Kernel model, RWCK Combined Rescorla-Wagner and Choice Kernel model, WSLS Win-Stay-Lose-Shift model.

corroborating that episodic simulations indeed induced such a preference ($W = 1225$, $p < 0.001$, $r = 0.87$, 95% CI = [0.80 Inf]; Shapiro Wilk: $W = 0.89$, $p < 0.001$; Fig. 2a, b).

**The simulation-induced value update generalizes to an external measure**

This preference shift is thought to reflect a positive update of how much the HR people are valued. If this is the case, we expected the value update to also be reflected in an external rating measure. Both

before and after the simulation task, participants indicated how much they liked the HR and LR people, as well as two further people from a baseline condition that they had not encountered during the simulation task. We could thus examine changes in liking induced by the simulations while controlling for any generic differences across the two measurements. On the initial test, people allocated to the three conditions neither differed in terms of how much they were liked ($F(2, 46) = 0.03$, $p = 0.97$, $n^2 < 0.01$) nor how familiar they were to the participants ($F(2, 46) = 0.21$, $p = 0.81$, $n^2 < 0.01$).

## Table 1 | Model comparison

| Models | LLmin | BIC | Number favoring | Exceedance probability | Model frequency |
|---|---|---|---|---|---|
| RW | 26.52 ± 2.5 1300 | 62.17 ± 5.1 3046 | 32 | 0.999 | 0.717 |
| RW-CK | 23.10 ± 2.3 1132 | 64.46 ± 4.6 3158 | 6 | 0 | 0.106 |
| CK | 31.95 ± 2.9 1566 | 73.03 ± 5.8 3578 | 11 | <0.001 | 0.168 |
| Noisy WSLS | 53.46 ± 0.8 2619 | 111.48 ± 1.6 5463 | 0 | 0 | 0.004 |
| Null | 65.2 ± 0.3 3195 | 134.96 ± 0.7 6613 | 0 | 0 | 0.004 |

Shown for each model: mean ± standard error of the mean and sum of minimum log-likelihood (LLmin); mean ± standard error of the mean and sum of the Bayesian Information Criteria (BIC); the number of subjects favoring each model based on BIC scores; exceedance probability; model frequency.

To examine the change in liking, we subtracted the liking ratings of the initial test from the one following the simulation task. We then corrected the change scores for the HR and LR conditions by subtracting the change scores of the baseline condition. This measure did not yield a significant change in liking for the LR people ($t(48) = 0.54$, $p = 0.59$, $d = 0.08$, 95% CI = [−0.01, 0.03]). By contrast, and consistent with our hypothesis, it revealed an increase in liking for the HR people ($W = 958$, $p < 0.001$, $r = 0.49$, 95% CI = [0.02, Inf]; Shapiro Wilk: $W = 0.94$, $p = 0.02$). This increase was more pronounced than the absent effect for the LR people ($W = 875$, $p = 0.004$, $r = 0.37$, 95% CI = [0.01, Inf]; Shapiro Wilk: $W = 0.95$, $p = 0.03$; Fig. 2c). This effect thus conceptually replicates and extends earlier work[18,19].

If the preference shift and the increase in liking are both manifestations of a simulation-induced value update, we further reasoned that these effects may be correlated with each other across participants. As a measure of acquired preference, we used the probability of choosing the HR people across the experimental session. As a measure of the increase in liking of the HR people, we corrected their liking update (i.e., the post–pre ratings) by subtracting the analogous update for the LR people. By this, both measures examine simulation-induced changes for the HR versus LR condition.

As predicted, these measures were indeed positively associated with each other, as assessed by a robust skipped Spearman's correlation ($r_s = 0.40$, 95% CI = [0.13, 0.63]; Fig. 2d). (Note that we also obtained a significant effect when performing this analysis with the uncorrected liking update for the HR people; skipped Spearman's correlation: $r_s = 0.38$, 95% CI = [0.11, 0.61]).

Overall, these results suggest that simulated experiences update our values in a manner similar to real experiences. In the next section, we examine the computational and neural processes that guide such learning.

### Reinforcement learning accounts for simulation-based learning

As detailed in the previous sections, episodic simulations induced a shift in preference for people who had been simulated more frequently in rewarding situations. We had hypothesized that such learning is based on an endogenous PE that drives RL. To examine this hypothesis, we tested whether the participants' trial-to-trial adaptations of their preferences can be captured by the RW model.

On each trial, the RW model updates the value of the chosen person as a function of the PE, scaled by the subject-specific learning rate $\alpha$ (a free parameter of the model). The PE, in turn, is calculated as the difference between the experienced reward, in this case the pleasantness of the imagined episode, and the current value of the person.

The values are then converted into the probability of selecting a person on a given trial, using a softmax decision rule that contains the freely estimated inverse temperature parameter, $\beta$. We first determined the individual best-fitting $\alpha$ and $\beta$ parameters, and then estimated the RW model's trial-wise choice probabilities and choice values. As evident in Fig. 2a, the model's estimated choice probability closely follows the trajectory of the participants' trial-wise choices.

Notably, across participants, a stronger overall probability of the RW model to select the HR people correlated with a greater increase in liking for the HR versus the LR people (see Supplementary Note 1). Much like the participants' actual choices, the choice probabilities of the model were thus associated with the external measure of the simulation-induced value update.

We further examined whether the RW model is better at accounting for the choices than a number of competing models of various complexities. Our alternative models explained choices as driven by (a) probabilistically sticking with the rewarded people while avoiding "punished" people (noisy win-stay-lose-shift; WSLS[35]), (b) repeating previous choices (choice kernel; CK[36]), (c) a combination of choice kernel and choice value (RW-CK[36]), and (d) random selection (Null) (for details, see Supplementary Methods).

For each model, we used the best-fitting parameter values to compute the Bayesian Information Criterion (BIC) as a measure of the model's goodness of fit. The simple RW model offered the best fit (Fig. 2e). We further corroborated this model's superiority using Bayesian model selection: Model frequencies and exceedance probabilities indicated that the RW model fits the data best of all the included models (Table 1).

As hypothesized, simulation-based learning was best captured by a model of RL that is driven by an endogenous PE.

### The decoded vividness of episodic simulations

Before turning to the neural implementation of the prediction error, we further scrutinized whether learning in our task was induced by the episodic simulations rather than merely evoked by the presented scenario cues. Towards this goal, we made use of a neural signature of prospective thoughts[37]. This whole-brain regression model allowed us to quantify, on a trial-by-trial basis, the experienced vividness of the simulated episode.

First, we observed that episodes were vividly simulated in both conditions as indicated by significant one-sample $t$ tests (HR: $t(48) = 11.6$, $p < 0.001$, $d = 1.66$, 95% bootstrapped CI [172.02, 240.98]; LR: $t(48) = 6.93$, $p < 0.001$, $d = 0.99$, 95% bootstrapped CI [108.35, 192.86]). These results indicate that participants likely performed the tasks as instructed. Importantly, the decoded vividness was stronger in the HR than in the LR conditions ($t(48) = 3.22$, $p = 0.001$, $d = 0.46$, 95% bootstrapped CI [22.74, 93.27]). The simulations were thus more vivid in the condition that also led to more learning, as indicated by the stronger liking update (Supplementary Fig. S6a).

Second, in the HR condition, episodes that were more vividly imagined were also experienced as more positive or negative (i.e., yielding a greater absolute pleasantness rating) ($t(3846) = 3.59$, $p < 0.001$, 95% CI [$8.58 \times 10^{-6}$, $2.92 \times 10^{-5}$]) (see Supplementary Fig. S6b). These analyses corroborate that learning occurred as a consequence of the internal simulation of *vivid* experiences.

### The endogenous PE is mediated by the ventral striatum

We then tested the hypothesis that the endogenous PE is mediated by activity in the ventral striatum. Specifically, we conducted a parametric modulation analysis to examine whether trial-by-trial variations in striatal activity can be accounted for by the model-derived time series of the PE. Given that the simulations of the episodes unfolded over the

## a  Striatal activity reflects PE

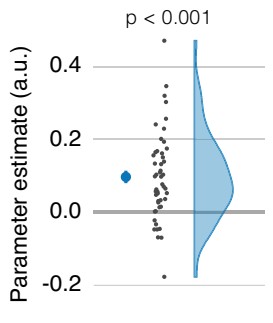

y = 14

**Parametric modulation by PE**

p < 0.001

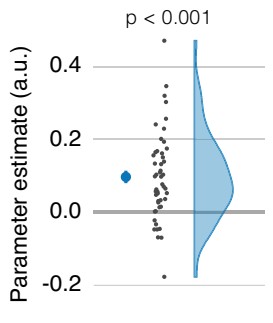

ventral striatum ROI

## b  dmPFC represents individual people and their value

Data (sketch)                    Prediction

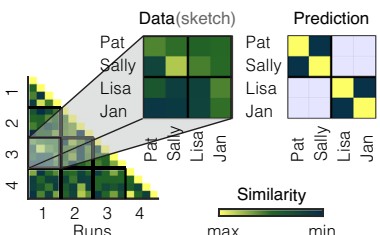

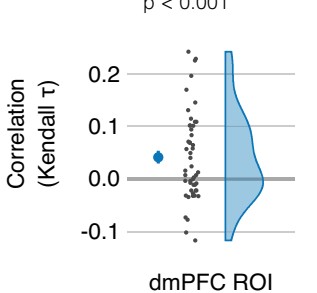

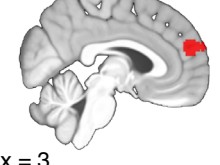

x = 3

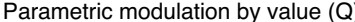

Similarity
max                    min

**Representational similarity analysis**
same-people vs. different-people similarity

p < 0.001

**Parametric modulation by value (Q)**

p = 0.008

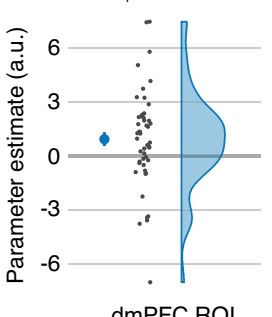

dmPFC ROI                              dmPFC ROI

**Fig. 3 | The neural basis of endogenous reinforcement learning. a** Activation in the ventral-striatal region-of-interest (ROI) was parametrically modulated by the trial-wise prediction error (PE) as estimated by the Rescorla-Wagner (RW) model ($p = 4e{-}07$). **b** A representational-similarity analysis indicated that the dorsomedial prefrontal cortex (dmPFC) encodes representation of individual people (left part of panel). This was reflected as more similar activity patterns during the simulation of the same people than of different people across functional runs. The plot depicts Kendall $\tau$ correlations between data-derived and predicted similarity matrices (top part of left panel) ($p = 5.6e{-}04$). Activity in the dmPFC was also parametrically modulated by the continuously updated value of the people (Q) as estimated by the RW model (right part of panel). Activity in the dmPFC thus carries information about the individual people and about their value. Larger dots denote the mean and the errors bars the standard error of mean; $n = 49$; Statistical inferences based on one-tailed one sample $t$ tests.

8 s of that task period, we modeled the activity with a boxcar regressor that covers the entire duration. This analysis was significant in our a priori region of interest (ROI), an anatomical mask of the ventral striatum (mask from Oxford-GSK-Imanova Structural-anatomical Striatal Atlas[38]) ($t(48) = 5.66$, $p < 0.001$, $d = 0.81$, 95% CI = [0.07, Inf]; Fig. 3a). Moreover, the activity in this area showed a poorer fit with an alternative model that codes for the simulation period as a transient event. Considering that this alternative model should be more sensitive to activity evoked by the external presentation of the scenario, the model comparison further suggests that the activity reflects the unfolding internal simulation (Supplementary Table S2).

We further corroborated these findings in a complementary whole-brain analysis. As predicted, this analysis yielded a cluster in the bilateral ventral striatum, specifically the nucleus accumbens, that extended into the anterior cingulate cortex (Supplementary Tables S3 and S4, and Fig. S7). We also obtained significant clusters in the bilateral anterior hippocampi and parahippocampal cortices, in the paracingulate cortex including the retrosplenial cortex, and the ventromedial prefrontal cortex (Supplementary Table S3 and Fig. S7).

### Striatal-dorsomedial prefrontal interactions support endogenous reinforcement learning

We next sought to test the hypothesis that the ventral striatum updates value by interacting with a cortical region involved in representing the information being updated. To examine this hypothesis, we first assessed whether the dmPFC encodes representations of individual people and of their value. We focused on this region, given its consistent involvement in thinking about other people[32,33].

### Dorsomedial prefrontal cortex codes for representations of people.

If the dmPFC encodes representations of individual people, we reasoned that the same representation should get reinstated whenever participants imagine an episode that features the same individual person. We examined this hypothesis by testing the replicability of activity patterns in the dmPFC using representational similarity analysis (RSA)[39,40].

Specifically, we employed an ROI defined by the Neurosynth meta-analysis for the term *people*. In this ROI, we examined whether the neural pattern similarity between episodes featuring the same person (same-person similarity) is greater than the neural pattern similarity between episodes featuring different persons (different-person similarity). To avoid any influence of the experimental condition on the results, the different-person similarity was only considered for the similarity of the respective two persons within a given reward condition. Consistent with our hypothesis, this effect was significant in the dmPFC ($t(48) = 3.47$, $p < 0.001$, $d = 0.50$, 95% CI = [0.02, Inf]; Fig. 3b, left panel).

We obtained consistent results in an exploratory whole-brain searchlight analysis (see Supplementary Note 4, Tables S7 and S8, and Fig. S9).

### Dorsomedial prefrontal cortex codes for the person's value.

The dmPFC thus seems to encode representations of individual people. Does this region also represent their value? To address this question, we took the model-derived value of the chosen person on any trial. This value, the variable Q, is continuously updated by the RW model based on the experienced PE.

We then examined whether regional activation during the choice period is parametrically modulated by Q. Consistent with our prediction, this effect was significant ($t(48) = 2.47$, $p = 0.008$, $d = 0.35$, 95% CI = [0.31, Inf]), indicating that activation in the dmPFC reflects the continuously updated value of the imagined person (Fig. 3b, right panel; see also Supplementary Tables S5 and S6, and Fig. S8). Together,

## Greater prediction errors are associated with stronger ventral striatal - dmPFC coupling

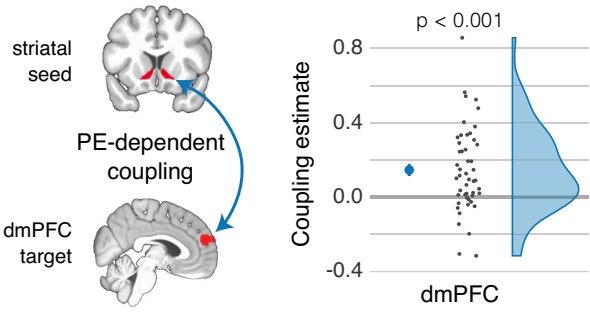

**Fig. 4 | Prediction-error dependent coupling between ventral striatum and dmPFC.** A psychophysiological interaction (PPI) analysis revealed that a greater prediction error (PE) is associated with a stronger functional connectivity between the ventral striatum and the dorsomedial prefrontal cortex (dmPFC). It is thus associated with a greater coupling between the region that elicits the value update and the region representing the information being updated. The figure displays the PPI effect in the dmPFC ($p = 2.2e\text{-}05$). The larger dot denotes the mean, and the error bar denotes the standard error of the mean; $n = 49$.

the results suggest that the people representations in the dmPFC also entail information about their value.

**Stronger striatal-dmPFC coupling accompanies stronger value updates.** We next examined whether the ventral striatum may elicit the value update by interacting with the dmPFC. Specifically, we conducted a psychophysiological interaction analysis that was seeded in the ventral striatum. The activation time-series was convolved with the trial-specific PE. The ensuing coupling parameters in the dmPFC were significant ($t(48) = 4.49$, $p < 0.001$, $d = 0.64$, 95% CI = [0.10, Inf]), indicating that, as predicted, a stronger value update coincided with a stronger functional connectivity between these regions (Fig. 4).

## Discussion

To ensure our survival, it is crucial that we continually learn from our past experiences and accurately predict and respond to changes in our environment. Our past experiences play a fundamental role in reinforcing our behavior, guiding us toward rewarding choices, and steering us away from choices that could lead to detrimental outcomes. Notably, our learning is not solely dependent on our direct experiences. We can also learn vicariously by observing the outcomes experienced by others[7,8,41].

In this study, we build on this seminal work and demonstrate that merely simulated rewards can induce learning and modify our real-life preferences. Episodic simulations have previously been shown to boost the retention of unpredicted information[42]. The current results go beyond these data in demonstrating that the PE itself can be a consequence of the simulated experience. The PE was calculated based on the experienced peasantness during the simulation, which, in turn, was stronger for more vivid episodes in the HR condition. Our findings indicate that this simulation-based learning is governed by the same computational and neural mechanisms that underlie direct and vicarious value-based learning.

This is remarkable for at least two reasons. First, simulation-based learning does not entail the actual receipt of a tangible primary (e.g., juice, shock) or secondary (e.g., money) reward. The rewarding outcome in the present study is the pleasantness experienced during the simulation. Yet, it can induce learning in the same fashion. Second, and relatedly, simulation-based learning differs fundamentally from experience-based learning in that the PE does not constitute a

mismatch between an internal model of the world and externally-derived information. Instead, the PE is based on a mismatch between our internal model and internally simulated information. Simulation-based learning is thus driven by an *endogenous* PE.

The endogenous PE seems to drive learning much like experience-based PE. Specifically, we found that this learning can be described as an effort to minimize these errors as formalized by the Rescorla-Wagner model. This simple model was better at accounting for the data than a number of alternative models of various complexities. The current data thus add to previous reports[18,19] that learning is not just driven by the frequency of simulations, but by the mentally experienced reward during those simulations and the ensuing PE. Over the years, the Rescorla-Wagner model has been shown to account for diverse phenomena—including Pavlovian and instrumental learning of simple features of the environment up to learning from complex social interactions[43,44]. Here, we generalize those findings by showing that there is no need for an actual reward to trigger learning.

The notion that simulation- and experience-based RL are based on a shared mechanism is further supported by our neuroimaging data. These highlight the contribution of the ventral striatum in mediating the endogenous PE. This region, and in particular the nucleus accumbens, has consistently been implicated as a key region for signaling PE, irrespective of whether they stem from direct or vicarious learning[28]. The present study further highlights the domain-general contribution of this region. In all of these cases, the ventral striatum performs the same computations—only the input that is driving these computations is different.

However, we do not suggest that it is just the striatum that mediates endogenous RL. Our exploratory whole-brain analysis identified a network of regions, including the ventral striatum, anterior hippocampus, and ventromedial prefrontal cortex. Activity in all of these regions is typically associated with the magnitude of a PE in neuroimaging studies. In particular, the hippocampus has been suggested to play a critical role by matching incoming information with internal representations. This process is suited for detecting novelty, which is then signaled to midbrain dopamine neurons via the ventral striatum[45]. Striatal[30,31] and, in particular, dopaminergic activity[24,46] then induce plasticity in cortical representations and thus afford learning. Notably, the hippocampus may support a dual role in the case of simulation-based learning. Given its critical involvement in the construction of a coherent scene[11,12], it may first foster the simulation of the very episode that is then being evaluated for relative novelty. This process may be mediated via big-loop recurrence[47], where the output of the simulation is relayed to the cortex before it then reenters the hippocampus for evaluation.

In this study, we showed that the dmPFC is involved in representing information about people[32–34] and their continuously updated value. This region showed a stronger coupling with the ventral striatum in the case of a stronger PE. We thus observed a learning-dependent interaction between the region eliciting the updating of information and the region involved in representing the information that is being updated. These interactions may be mediated by established anatomical connections between these regions, for example, through the cortico-basal ganglia-thalamo-cortical circuit[48].

The observation that PE can update our attitudes toward personally familiar people is reminiscent of data demonstrating that a mnemonic prediction error can also update episodic memories[49–52]. Specifically, PE can disrupt sustained representations of episodic memories, rendering them malleable and allowing for the integration of new information[49]. These data align with the current results in showing that PE can lead to the modification of declarative long-term memory representations.

An important avenue for future research will be to examine how features of the imagined episode influence the ensuing learning. For example, in the majority of the scenarios, the imagined person did not

cause the specific event. Learning may not only be stronger but also qualitatively different if the scenarios were to imply such causality or intentionality[53–55]. Moreover, we observed that more vivid simulations (as decoded from brain activity) were associated with stronger affective experiences in the HR condition. This suggests that learning could potentially be boosted by enhancing the vividness of the simulation. This could be achieved, for example, by manipulating the familiarity of the locations at which the episodes are being imagined[56,57].

Another avenue will be to further probe whether the described mechanism of simulation-based learning can also lead to a devaluation, i.e., a decrease in liking. In the current study, persons in the LR condition did not become more disliked at the end of the session, although they experienced a number of unpleasant-to-neutral episodes. This absence of evidence may suggest that simulation-based learning can only lead to an upward shift in value. However, we believe that it may be more accurately attributed to other factors. First, simulations in the LR condition were less vivid and thus likely induced a lower experienced reward as a driver of learning. Another possibility is the valence of the provided scenarios. Whereas the majority of the scenarios in the HR condition were positive, they ranged more widely, from unpleasant to neutral, in the LR condition (Supplementary Fig. S1). A selection of more negative scenarios might also yield a more negative shift in liking. Third, we have previously shown that simulation-based learning can be based on two complementary mechanisms[18]: a more specific one that relies on the valence of the imagined episode and a generic one that, akin to mere exposure, renders the contents of repeatedly imagined events more positive– irrespective of the valence of the episode. We suggest that this latter effect may have offset any of the former effects in the LR condition.

The demonstration of endogenous RL adds to the accumulating evidence that episodic simulation allows us to bootstrap from our experiences to forge new insights. This capacity has both beneficial and unwelcome consequences. On the one hand, episodic simulation constitutes a powerful learning mechanism that does not depend on any new experiences. For example, simulations can help us to come up with solutions to dreaded situations and thus mitigate our apprehensiveness about the future[20] (see also ref. [58]). Moreover, simulations allow us to mentally experience how we would feel in prospective episodes[32,59], and this experience can motivate more farsighted decisions[60,61].

On the other hand, there is also a downside to a learning mechanism that is decoupled from environmental feedback. In particular, individuals with elevated anxiety or depression show a negative affective bias and have a higher learning rate for negative outcomes[62]. There is also some preliminary evidence that people high in neuroticism learn less from imagined positive experiences[18]. Without correcting feedback, simulation-based learning may thus contribute to the maintenance of various affective disorders[63,64]. Sometimes it may thus be more beneficial for one's well-being to stop simulating hypothetical events[58,65,66].

To conclude, the present study reveals fundamental principles of how we learn from merely imagined experiences. These experiences contrast with our internal expectations and thus induce an endogenous PE. Our simulations thus feed into a more general learning mechanism that seeks to reduce uncertainty and that is mediated, among other regions, by the ventral striatum and its interactions with the neocortex. A better understanding of this mechanism may elucidate the origin of a number of maladaptive psychological processes and allow us to harness its adaptive potential. More broadly, it will be integral for our comprehension of how we create models of our world.

## Methods
### Participants
We recruited 50 participants with no reported history of neurological and psychiatric disorders for the study. The sample size was determined based on the effect size ($d = 0.69$) reported in a previous study on simulation-based learning[19]. It provides 90% power for the critical paired $t$ tests comparing the HR and LR conditions (at $\alpha = 0.001$, one-tailed). The study was approved by the ethics committee at Leipzig University (ethics number 122/19-ek). All participants provided written informed consent prior to participation. Participants were reimbursed at a rate of €9 per hour for the behavioral part and €10 per hour for the MRI part of the study. One participant fell asleep during the MRI session and was thus excluded from analyses. We accordingly included data from 49 participants (24 female, 25 male; age: $M = 27.6$ y; SD = 4.8 y, range = 19–35 y). Prior to data collection, we validated our stimulus material on an independent sample ($n = 107$; 57 female, 50 male; age: $M = 24.1$ y; SD = 3.8 y, range = 19–35 y) (see below).

### Materials
This study examined whether episodic simulations induce learning about personally familiar people. Similar to refs. 18,19, we thus asked participants to compile a list of 90 names of people that they know from their everyday lives (Fig. 1a). They then indicated how much they liked the people. We take liking as a proxy for the degree to which they value a given person. We also assessed how familiar they were with each person. Note that we only acquired these ratings for the last 30 people that the participants had provided. This is because people listed earlier tend to be those who are particularly liked, whereas we were interested in selecting more neutral acquaintances that tend to be listed later[18].

Specifically, the participants placed the names of the 30 people on a rectangular space with rating values on the y-axis (Fig. 1b). The y-axis ranged from "not at all liked" to "very much liked" and from "not at all known" to "very well known". This measurement allowed us to assess the relative liking and familiarity of the people in a fine-grained fashion.

Based on the ratings, we selected six people who were neutrally liked and also sufficiently familiar to the participants. Two of these people were assigned to the HR, LR, and baseline conditions, while we aimed to minimize differences in liking and familiarity across conditions.

With an independent sample of participants ($n = 107$), we validated a set of 129 sentences describing scenarios that are either pleasant (67) or neutral-to-unpleasant (62) and that can be imagined in detail (see Supplementary Methods for details). The sentences served to induce the simulations of specific episodes in the MRI scanner. All scenarios along with their pleasantness ratings can be found on the project's Open Science Framework repository at https://doi.org/10.17605/OSF.IO/Q3V6B.

### Experimental procedure
In the MRI scanner, participants performed a simulation task that is akin to a two-armed bandit task (Fig. 1c). Each trial started with a choice phase. Here, they were presented with the names of an HR and an LR person on either side of a fixation cross for a maximum of 3 s. Participants aimed to select the person with whom they would more likely imagine a pleasant experience in the next step.

Once participants had made their choice, only the selected person remained on screen next to the fixation cross for 2.5 s. This was followed by only the fixation cross for 2.5 s plus any time remaining from the choice phase. Afterward, participants saw the chosen name together with a sentence describing a naturalistic scenario on screen for 8 s. During this period, they imagined interacting with the person in the presented scenario.

The people in the HR condition were imagined in pleasant scenarios with a higher probability than those in the LR condition (80% vs. 30% of the trials). In the remaining trials, participants imagined interacting with the selected person in one of the neutral-to-unpleasant scenarios.

The specific scenario was randomly chosen for any given trial. Towards the end of the 96 trials of a session, we had to repeat a limited number of scenarios in a number of participants (pleasant events in 29 participants: mean number of repetitions = 4.48, range 1–7; neutral-to-unpleasant events in 6 participants, mean = 1, range = 1–1). In some of those few cases, the same scenario was imagined with two different people.

At the end of each trial, participants indicated the pleasantness of the imagined episode on a continuous slider scale, ranging from "very unpleasant" to "very pleasant" with a "neutral" mid-point, within a maximum of 5 s. We use this outcome measure as a proxy for the experienced reward and use it to model the PE on a given trial. The trial then concluded with a fixation cross during the ITI that was presented for a jittered period ranging from 2–5 s ($M = 3.3$ s, SD = 0.83 s) plus the remainder of the time from the preceding rating phase. Overall, the task consisted of 96 trials that were split into four blocks. For another project, the participants also performed a standard bandit task (not reported here).

Outside the scanner, participants once again rated the 30 people on the liking and familiarity scales (post-task ratings). They then completed tasks designed to assess their memory for the simulation task and four questionnaires (a short version of the Big Five Inventory, Beck Depression Inventory II, Mind Wandering Questionnaire, and Vividness of Visual Imagery Questionnaire). The results from the memory tasks and the questionnaires are not reported in this manuscript. To conclude the session, participants were debriefed about the purpose of the study and monetarily reimbursed for their time.

## Computational models

We fitted five models to participants' trial-wise choices during the simulation task. The model space included (a) a standard RL model that captures learning as driven by PEs, i.e., the Rescorla-Wagner (RW) model; (b) a model that captures the propensity of merely repeating previous choices (Choice Kernel model; CK); (c) a combination of the RW and CK models (RW-CK model); (d) a model that captures the tendency to repeat rewarded actions while shifting away from unrewarded actions (Win-Stay-Lose-Shift; WSLS); and (e) a model that captures no learning (Null model). Below, we provide details of the RW model. Details of the other four models are described in the Supplementary Methods.

The RW model updates the expected value ($Q$) of the selected choice ($k$) based on the reward ($r$) received on a given trial ($t$). In our simulation task, the familiar person chosen on a given trial $t$ constitutes the choice $k$. We binarized the rated pleasantness and used it as a proxy for the reward $r$. The learning rule was applied as follows:

$$Q^k_{t+1} = Q^k_t + \alpha \delta_t \tag{1}$$

$$\delta_t = r_t - Q^k_t \tag{2}$$

where $\alpha$ is the learning rate ranging between 0 and 1 that determines the extent to which the PE $\delta_t$ drives the update of the expected value $Q^k_{t+1}$. We initialized the choice value ($Q^k_{t=0}$) at 0.5, given that the imagined people were considered to be approximately neutral before the start of the task.

We used the softmax choice rule to determine the likelihood that a person is selected on the subsequent trial. The rule uses the computed value of the person as follows:

$$p^k_t = \frac{\exp(\beta Q^k_t)}{\sum_{i=1}^{K} \exp(\beta Q^i_t)} \tag{3}$$

where $\beta$ is the inverse temperature parameter controlling for randomness in choice. It ranges from 0 (completely random) to $\infty$ (deterministically choosing the most valued person). Overall, the RW model thus has the two free parameters $\alpha$ and $\beta$.

## Model comparison

We sought to assess whether the RW model accounts best for the empirical data by comparing the fits of the different models. To this end, we first fitted each model to participants' data by optimizing the model parameters. This was done by minimizing the negative log-likelihood using Matlab's fmincon function.

Using the negative log-likelihoods (NegLL), we calculated the BIC for each model and each participant as follows:

$$BIC = \log(n)df + 2NegLL \tag{4}$$

where $df$ refers to the number of free parameters and $n$ refers to the number of data points. BIC penalizes the maximum likelihood by the number of free parameters. We computed the inter-individual difference in the BIC of all models in relation to the BIC of our hypothesized model, i.e., the RW model ($\Delta BIC = BIC_{RW} - BIC_{model}$) (Fig. 2e).

To assess whether a model fits the data better than all other models in the model set, we analyzed group-level model fit by estimating the expected frequency and exceedance probability of the model using the VBA toolbox[67] (see Table 1).

Before setting up these analyses on the empirical data, we first ensured that the free parameters of our models could be accurately recovered in simulated data (see Supplementary Methods and Fig. S2). Moreover, we performed a model recovery analysis based on the simulated data to ensure that we can arbitrate different models (see Supplementary Methods and Fig. S3). The analyses were conducted as outlined in ref. 36.

## fMRI data acquisition

Participants were scanned with a 3 Tesla Siemens SKYRA scanner with a 32-channel head coil at the Max Planck Institute for Human Cognitive and Brain Sciences. We acquired structural images with a T1-weighted (T1w) MPRAGE protocol (176 sagittal slices with interleaved acquisition, field of view = 256 mm, 1 mm isotropic voxels, TR = 2300 ms, TE = 5.28 ms, flip angle = 9°, phase encoding: anterior-posterior). For each of the four functional runs, we acquired 307 volumes of blood-oxygen-level-dependent (BOLD) data using a whole-brain multiband echo-planar imaging (EPI) sequence (field of view = 204 mm, 2.5 mm isotropic voxels, 60 slices with interleaved acquisition and MF = 3, TR = 2000 ms, TE = 22 ms, flip angle = 80°, phase encoding: anterior–posterior). The first four volumes of each run were discarded to allow for T1w equilibration effects.

## fMRI preprocessing

The MRI data were converted to the Brain Imaging Data Structure[68]. The imaging data were then preprocessed using the default preprocessing steps of fMRIprep version 20.2.6 based on Nipype 1.7.0[69].

The T1w image was corrected for intensity non-uniformity and used as a reference throughout the workflow. The T1w-reference was then skull-tripped and segmented into cerebrospinal fluid (CSF), white matter (WM), and gray matter. It was then normalized to MNI space (MNI152NLin2009cAsym).

The functional imaging data were corrected for slice timing using 0.5 of the slice acquisition range, head motion using the estimated transformation matrices, and the six rotation and translation parameters. They were also corrected for susceptibility distortions using the fieldmap acquired prior to functional MRI imaging. The BOLD reference was co-registered with the T1w reference using boundary-based registration and configured with six degrees of freedom. Several

confounding time series were calculated based on the preprocessed BOLD, including the framewise displacement (FD). Additionally, a set of physiological regressors was extracted to allow for component-based noise correction (CompCor). For anatomical CompCor, three probabilistic masks (CSF, WM, and combined CSF+WM) were generated in anatomical space. For further details, please refer to the online documentation (https://fmriprep.org/en/20.2.6/).

The parametric modulation and effective connectivity analyses were performed in MNI space after smoothing with a Gaussian kernel of 6 mm FWHM. The representational similarity analyses were performed in native space.

## Parametric modulation analysis

The functional MRI data were further analyzed using SPM12 (https://www.fil.ion.ucl.ac.uk/spm/). We performed a parametric modulation analysis to assess activation differences associated with trial-by-trial variations in PE and choice value. Specifically, we estimated a general linear model (GLM) with a boxcar regressor that coded for the 8 s of the simulation period for each trial on which participants had made a choice. This GLM also included regressors coding for the onset of the choice presentations and of the ratings with stick functions. If a subject had missed at least a single trial, we included an additional regressor modeled with a stick function that coded for the onset of each missed trial. Importantly, we included two parametric modulators derived from the RW model: the trial-by-trial PE ($\delta_t$; Eq. 2) modulating the activation during the simulation period and the estimated choice value ($Q_t^k$; Eq. 1) during the choice presentation. All of these regressors were convolved with the canonical hemodynamic response function.

As recommended by ref. 70, we used the population median of the $\alpha$ and $\beta$ parameters to estimate the trial-wise PE and Q, given that unregularized random-effects parameter estimates, such as the subject-specific $\alpha$ and $\beta$ parameter estimates, tend to be too noisy to obtain reliable neural results.

In our procedure, participants continue to learn the choice value across the whole session. We therefore performed the parametric modulation analysis on data that were concatenated across the four functional runs. As nuisance regressors, we further included the six head motion parameters, FD, the first six CompCor components for each run, and block regressor for the concatenated runs. We adjusted the high-pass filter and temporal non-sphericity calculations to account for the four independent functional runs as implemented in SPM12.

For our ROI analyses, we took the mask of the ventral striatum from the FSL Oxford-GSK-Imanova Structural–anatomical Striatal Atlas[38] and defined dmPFC based on regional cluster obtained from the Neurosynth meta-analysis for the term *people* (association test map). For a given analysis, we then averaged the parameter effect for the respective effect from all voxels within these ROI.

Additionally, we carried out complementary exploratory whole-brain analyses by entering the respective contrast estimates into a second-level analysis. The results from the whole-brain analyses are reported in the Supplementary Information.

## Representational similarity analysis

We used RSA to assess whether the dmPFC region encodes representations of individual people. This analysis used functions from the RSA toolbox[71]. We first estimated a GLM that included one boxcar regressor for each imagined person and functional run. These regressors coded for the 8 s of the simulation periods but only for those trials that were congruent to the respective person's reward category. That is, for HR people, it only included trials with pleasant scenarios; for LR people, it only included trials with neutral-to-unpleasant scenarios.

We then quantified the degree to which neural pattern similarity in the dmPFC was consistent with our prediction, namely that it codes

for representations of individual people (Fig. 3B, left panel, top). In a first step, we created a 16 × 16 model representational similarity matrix (model RSM) containing predicted pairwise similarities between the 16 neural patterns (4 persons x 4 fMRI runs). Same-person, different-run pairs were coded as 1; different-person, different-run pairs were coded as −1. Because same-person pairs are always from the same reward condition, we likewise restricted the different-person pairs to pairs from the same reward condition. This was done to ensure that any difference with the same-person similarity is not a result of general condition differences (in particular, those related to the value of the imagined event).

In the next step, we computed the similarity in activity patterns between all pairs of persons and runs across all voxels of the dmPFC ROI, using Kendall $\tau$[71]. This likewise resulted in a 16 × 16 neural RSM. Lastly, for each participant, we computed Kendall $\tau$ correlations between the model and neural RSMs. Correlation coefficients greater than zero thus indicate that this ROI represents episodes featuring the same person as more similar than episodes featuring different persons. We complemented this anatomical RSA analysis with a spatially unconstrained whole-brain searchlight RSA (see Supplementary Note 4).

## Psychophysiological connectivity analysis

We computed a psychophysiological interaction (PPI) analysis to test whether the functional connectivity between the ventral striatum (seed region) and dmPFC (target region) varies as a function of the PE. This analysis was based on the GLM from the PE parametric-modulation analysis. The physiological regressor comprised the first eigenvariate of the activity within the ventral striatum, adjusted for effects of interest. The psychological regressor comprised the trial-by-trial PE ($\delta_t$; Eq. 2), estimated using the fixed median $\alpha$ and $\beta$ parameters. We computed the PPI regressor by convolving these two regressors, i.e., the activation of the ventral striatum and estimated trial-wise PE. The GLM included these regressors and was estimated with adjusted high-pass filter and temporal non-sphericity calculations to account for the original runs in the concatenated design matrix. To examine changes in coupling with the dmPFC, we extracted the ensuing parameter estimates from our mask for this region.

## Vividness classifier

We unobtrusively assessed the vividness of the individual episodic simulations by using a neural signature of the vividness of prospective thoughts[37]. This signature is a whole-brain least-squares regression weight map that provides an estimate of the experienced vividness based on the observed brain activity on a given trial. We obtained the classifier upon personal request from the first author. It is also openly available at https://www.thedecodinglab.com/resources.

We first resampled and registered the preprocessed data to the standard fMRIprep MNI 2-mm template and, thus, to the original signature map. We then calculated a separate GLM for each trial[72,73]. Each model comprised one regressor coding for a respective trial and another regressor coding for all other trials. As in the other analyses, the regressors covered the 8 s of the simulation, and motion parameters were included as regressors of no interest. We concatenated the data across the four functional runs, and accordingly adjusted the high-pass filter and temporal non-sphericity estimates as implemented in SPM12.

We then smoothed the resulting parameter maps with a Gaussian kernel of 8 mm FWHM as in ref. 37, and calculated the dot product of the parameter maps and the signature map. These dot product values were aggregated by condition (HR vs. LR) and participant, and submitted to a paired-samples $t$ test. We also ran a linear mixed-effects model with decoded vividness and condition (HR vs. LR) as fixed effects and the participant as a random effect to predict absolute pleasantness across individual trials. We followed up on the significant

interaction in this analysis with two separate models for the HR and LR conditions (see also Supplementary Fig. S6).

## Statistical analysis

Statistical tests were conducted with R version 4.4.1 (www.r-project.org). All directed predictions were examined with one-tailed tests. Whenever the Shapiro-Wilk test indicated violations of normality, we used Wilcoxon Signed Rank Test in place of the Student's $t$ test. The skipped Spearman's correlations were computed using the robust correlation toolbox[74] implemented in Matlab (MATLAB version 9.10, R2021a).

## Data availability

Pseudonymized behavioral data and a list of scenarios have been deposited in the Open Science Framework under https://doi.org/10.17605/OSF.IO/Q3V6B. The second-level t-maps for the parametric-modulation analyses based on PE and Q, searchlight RSA, and psychophysiological interaction analyses, as well as anatomical masks of the dmPFC and ventral striatum ROI, have been deposited at Neurovault under https://identifiers.org/neurovault.collection:17353. The raw MRI images are available under restricted access due to identifiable personal markers. Access can be obtained by contacting the corresponding authors.

## Code availability

All custom code for our behavioral, computational, and fMRI analyses has been deposited in the Open Science Framework under https://doi.org/10.17605/OSF.IO/Q3V6B. The code for the computational models is based on previously published research[36,67].

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

## Acknowledgements

We thank the Max Planck Society for supporting this work through a Max Planck Research Group awarded to R.G.B. A.D. acknowledges partial support from the Deutscher Akademischer Austauschdienst and the Max Planck School of Cognition. R.B. acknowledges support from the Deutsche Forschungsgemeinschaft (grant no. 412917403). We thank Maria Woitow and Roxanne Eisenbeis for assistance in developing and testing the scenarios, and Marie-Louise Iredale, Charlotte Gmeiner, Marie-Kristin Mueller, Lena Pfund, Lena Kuschel, and Brais Gonzalez Sousa for their support with participant recruitment and data collection.

## Author contributions

A.D. and R.G.B. conceptualized and designed the study, A.D. collected the data, A.D., R.B., H.S., F.B. and R.G.B. contributed to the analyses, A.D. conducted the formal analysis, A.D. and R.G.B. wrote the original paper, and all five contributed to reviewing and editing the paper.

## Funding

## Competing interests

The authors declare no competing interests.
