## [Transparent Peer Review file · Nature Communications]

Learning from imagined experiences via an endogenous prediction error

Corresponding Author: Ms Aroma Dabas

Version 0:

Reviewer comments:

Reviewer #1

(Remarks to the Author)

This study investigates the interesting idea that reinforcement learning based on unexpected reward signals can also happen based on internal events that we have merely imagined. To test this, an imagery version of the stable two-armed bandit task was developed in which participants were asked to imagine pleasant, neutral or unpleasant scenarios involving the person they chose on any given trial. If similar learning processes would be at play as during perceived reward, participants should show behavioural and neural signatures in line with model-updating over the course of the task. Behaviourally, participants indeed update their value preferences based on these imagined scenarios, which was captured by a RW model. Furthermore, model-based PE during the task was captured by fluctuations in striatal activity which was in turn coupled to dmPFC, which contained representations of the individual people. The study is well-executed, well-written and original. However, I have a number of comments that I feel need to be addressed. Most importantly, I am not sure whether this study investigates imagination.

Major comments:

The paper consistently talks about 'imagined', 'simulated' or 'endogenous' experience/signals. However, how can we be sure that the effects are due to imagination/simulation rather than just reading the positive/negative text on the screen? The novel information is externally presented to the participant on the screen. Therefore, why would this be an imagination-based learning signal? To test whether imagining had any influence, either a measure of imagery strength/success needs to be added (e.g. vividness), that then shows an influence on the model updates, or a control condition in which participants just read the text and do not imagine anything is needed.

Line 243, as far as I understood the experimental design correctly, the same person was always associated with LR or HR (hence 'stable' two-armed bandit). How then is it possible to calculate representations of different people within LR and HR conditions separately?

Minor comments:

On line 147, the reported analysis seems to be a convoluted way of performing a repeated measures ANOVA with time and reward condition as within-subject variables. Why is this alternative approach used? Furthermore, if models are being updated based on reward, why wouldn't this also be reflected in a decrease in liking for the LR people?

Missing p-values for Spearman correlations on line 162.

Reviewer #2

(Remarks to the Author)

Dabas et al. adapted the logic of two-armed bandit tasks to a "social imagination" task. On each trial, participants were asked to choose between persons that they knew and had previously rated as rather neutral. Participants were then asked to imagine interacting with the chosen persons in more or less pleasant scenarios. Crucially, the probability of a person being linked to pleasant scenarios differed such that they were designed to entail high or low reward. For these imagination phases, participants learned to choose "high reward persons" more often than "low reward persons," which was reflected in

overall changes of linking ratings (pre vs. post the learning task) and in a standard simple Rescorla Wagner model. The prediction errors from this winning model were related to fMRI signals in anatomically defined ROIs of the striatum. As shown by RSA, the to-be-imagined persons were related to fMRI signals in a part of the dmPFC (defined via neurosynth). Signals in this region were related to the Q value of the persons derived from the Rescorla Wagner model. PPI showed a coupling between the two regions.

In our view, this is a very clever and elegant study that shows how imagined social interactions with other persons can drive the evaluation of these persons. Analyses seem to be well-conducted and rather comprehensive given the study design. We could not detect any obvious flaws in the methodological approach, which is state-of-the-art in this field. The paper is very clearly written.

Here, a couple of questions and comments:

1. Involved brain regions:

- a. We were surprised that the authors did not mention parts of the MTL when motivating their design in the introduction. They do find hippocampal activity in whole brain analyses but do not show maps and only very briefly mention this in the discussion. Could you please expand this a bit more or explicitly mention why you do not focus on this region?
- b. Overall brain maps should be shown for the whole brain analyses and discussed in a bit more detail. Tables are more difficult to parse.
- c. In the figure headings, it is not quite clear that the shown regions are ROIs (at least for the dmPFC).

2. RSA:

- a. The authors write: "To avoid any influence of the experimental condition on the results, we calculated the different-person similarity separately for the HR and LR conditions." This is nice but we did not understand how the authors then calculated the overall pattern. Were the two conditions merged? Did they differ?
- b. It is not quite clear how many events were used. Were some of them repeated? Wouldn't it be possible to run RSA on the similarity (in pleasantness) of the events? Would you expect this similarity to be captured in the same ROI? ... maybe along with the general pleasantness ratings?

3. Events: In general, some more details would be nice in the main texts and especially in the SI.

- a. It would be important to have a list of the events along with the ratings of these from the two samples.
- b. What was the variability in ratings? Some scenarios might be more positive/negative (e.g., depending on whether you like ice cream)
- c. Can the events be categorized into more or less mundane, frequent, special events? E.g., eating ice can be imagined with any person but going to the opera or a metal concert might be difficult to imagine with same people. Relatedly, does the difficulty/oddness/vividness of the imagination have an influence on how "rewarding" it is? As far as we see, the authors did not specifically assess this but 1-2 lines of discussion would be good.
- d. Related to the previous point, are there often causal links between event and person implied (e.g., Sally has broken your bike or she just has found it? Has Harry invited you to eating ice cream or have you just met him coincidentally?). Again, 1-2 lines of discussion could be interesting.
- e. Did all participants see the very same events? How was the order randomized? Did they see some events twice? ... with different persons? See comment above for RSA.

4. Persons:

- a. Did the imagination also change the rated familiarity with these persons?
- b. One of the crucial points related to the previous point that we did not quite understand is: Since participants learned to choose the HR persons more often than the LR persons, did they also get to imagine more scenarios with the HR persons? So, to what degree are effects driven by the frequency of imagination versus the pleasantness of the imagined scenario? What would happen if participants were "forced" to imagine each person with the same frequency but with more or less pleasant events. Instead of an explicit choice, participants could have given a rating to fit the model. A few lines of discussion would be interesting.

5. Did the authors preregister parts of the study?

6. Parameter recovery in Fig. S2. What was the range of tested parameters?

7. Figure 2 "simulations:" This word is confusing because it sounds as if the data were simulated. Imagination might be better.

Reviewer #3

(Remarks to the Author)

Version 1:

Reviewer comments:

Reviewer #1

(Remarks to the Author)

I thank the authors for revising their manuscript in response to my previous comments.

In response to my previous concern that the activations might reflect a response to merely reading the presented text rather than vividly imagining the suggested scenarios, the authors have now added two extra control analyses. The first analysis

looks at the dynamics of the neural activations, based on the assumption that if the activation reflects stimulus processing instead of imagination, it would be short lived. The second analysis quantified the experienced vividness of imagination using a 'vividness-classifier' on the whole-brain activity and showed that decoded vividness was higher in the high-reward compared to the low-reward condition. These are both excellent control analyses.

However, it was not clear to me how the vividness classifier was defined, given that no vividness ratings were collected in this data set. Did the authors use the classifier that was developed by Lee et al. (2022)? Looking at that paper, I did not find a link to the whole-brain regression weights, but I might have missed it. More explanation on how this analysis was executed is needed to evaluate the validity.

Furthermore, I appreciate that the authors demonstrate that similar statistical conclusions can be obtained with a more standard repeated-measures ANOVA. I do not have a strong preference for including one analysis over the other, my only thought is that the ANOVA might be more straightforward for a lot of readers to interpret.

Besides these, all my other comments have been addressed, and I would like to congratulate the authors on an excellent piece of work.

(Remarks on code availability)

The code is well-documented and clear.

Reviewer #2

(Remarks to the Author)

Thank you very much for addressing my and my co-reviewer's comments. We are confident that this paper will be read widely and will inspire many new studies.

(Remarks on code availability)

Reviewer #3

(Remarks to the Author)

(Remarks on code availability)

Version 2:

Reviewer comments:

Reviewer #1

(Remarks to the Author)

The authors have address all my comments.

(Remarks on code availability)

We thank the editor and the three reviewers for their helpful feedback and suggestions. As detailed below, we have carefully addressed each raised point in the revised manuscript. In particular, we have considerably strengthened our inference that the observed learning is based on the vivid simulation of the respective episodes.

Start of reply to reviewer comments

Reviewer #1 (Remarks to the Author):

This study investigates the interesting idea that reinforcement learning based on unexpected reward signals can also happen based on internal events that we have merely imagined. To test this, an imagery version of the stable two-armed bandit task was developed in which participants were asked to imagine pleasant, neutral or unpleasant scenarios involving the person they chose on any given trial. If similar learning processes would be at play as during perceived reward, participants should show behavioural and neural signatures in line with model-updating over the course of the task. Behaviourally, participants indeed update their value preferences based on these imagined scenarios, which was captured by a RW model. Furthermore, model-based PE during the task was captured by fluctuations in striatal activity which was in turn coupled to dmPFC, which contained representations of the individual people. The study is well-executed, well-written and original. However, I have a number of comments that I feel need to be addressed. Most importantly, I am not sure whether this study investigates imagination.

Major comments:

Comment 1:

The paper consistently talks about 'imagined', 'simulated' or 'endogenous' experience/signals. However, how can we be sure that the effects are due to imagination/simulation rather than just reading the positive/negative text on the screen? The novel information is externally presented to the participant on the screen. Therefore, why would this be an imagination-based learning signal? To test whether imagining had any influence, either a measure of imagery strength/success needs to be added (e.g. vividness), that then shows an influence on the model updates, or a control condition in which participants just read the text and do not imagine anything is needed.

Our reply: We thank the reviewer for raising this point, which we agree is fundamental to the interpretation of this study. In response, we have conducted two complementary sets of analyses that bolster our argument that the observed effects reflect the vivid imagination of prospective episodes.

First, we further scrutinized the activity in the ventral striatum associated with the prediction error. We reasoned that any brain activity that is primarily due to the external presentation of the prompts should be evoked by the cue and then be short-lived. As such, the associated prediction error should correlate with more transient activity that could best be modeled with an impulse function (i.e., with a duration of 0s). By contrast, any activity reflecting the imagination of an episode should unfold over the course of the simulation period. It should thus be better captured by a model that includes a boxcar regressor that accounts for sustained activation over those 8s. Indeed, this is why we opted to model brain activity in the latter fashion. In the revised manuscript, we make a formal comparison between the two options (i.e., modelling the prediction error with an impulse versus a boxcar function). Bayesian Model Selection clearly indicates that the boxcar function is superior at accounting for the data, supporting

the hypothesis that the prediction error arises internally as a consequence of the unfolding simulation. We refer to this analysis in the revised manuscript (p. 8-9) and report it in detail in the supplement (section 6 and Table S2).

Second, we also addressed this issue by following the reviewer's advice to examine whether vividness, as a measure of imagery strength, influenced learning. Due to the lab's recent move from Germany to the US, and the lead author's move into industry, we were not able to run another behavioral study in this short time frame. However, the fMRI data allowed us to establish a clear link between vividness and learning. Specifically, we made use of a neural signature of the vividness of future simulations that was recently developed by Lee et al., (2022). Specifically, based on the data reported in Lee et al., (2021), they trained a decoder to unintrusively quantify the vividness of future simulations from associated brain activity. The authors then further validated this classifier in showing that it can differentiate experimental conditions that should vary in terms of vividness.

Here, we made use of this signature to decode vividness of episodic simulations on a trial-by-trial basis (see revised methods for more details; p. 20). As described in the revised results, we use this approach to provide two major pieces of evidence for our hypothesis. On the one hand, we show that the episodes were more vividly simulated in the high-reward than the low-reward condition – i.e., in the condition that was also associated with stronger learning. On the other hand, we show that episodes that are more vividly imagined are also experienced with a stronger affect, i.e., as more pleasant and more unpleasant, respectively. These data thus clearly indicate that the success of the simulation was related to the pleasantness, our marker of the experienced reward. This association was present in the high reward condition that also led to a stronger value update.

Together, the two sets of analyses considerably strengthen our interpretation that learning occurred as a consequence of the internal simulation of vivid experiences.

Action taken: We have described the additional analyses in the revised methods (p. 20), results (p. 8-9), and supplement (p. 9-10) as well as referred to in the discussion (p. 11, 13).

Results from p. 8-9:

The decoded vividness of episodic simulations

Before turning to the neural implementation of the prediction-error, we further scrutinized whether learning in our task was induced by the episodic simulations rather than merely evoked by the presented scenario cues. Towards this goal, we made use of a neural signature of prospective thoughts³⁷. This whole-brain regression model allowed us to quantify, on a trial-by-trial basis, the experienced vividness of the simulated episode.

First, we observed that episodes were vividly simulated in both conditions as indicated by significant one-sample t-tests (HR: $t_{48} = 11.6$, $p < 0.001$, $d = 1.66$; LR: $t_{48} = 6.93$, $p < 0.001$, $d = 0.99$). These results indicate that participants likely performed the tasks as instructed.

Importantly, the decoded vividness was stronger in the HR than in the LR conditions ($t_{48} = 3.22$, $p = 0.001$, $d = 0.46$). The simulations were thus more vivid in the condition that also led to more learning as indicated by the stronger liking update (Supplement Fig. S6a). Second, in the HR condition, episodes that were more vividly imagined were also experienced as more positive or negative (i.e., yielding a greater absolute pleasantness rating) ($t_{3846} = 3.59$, $p < 0.001$) (for details see Supplement Fig. S6b). These analyses corroborate that learning occurred as a consequence of the internal simulation of *vivid* experiences.

The endogenous PE is mediated by the ventral striatum

We then tested the hypothesis that the endogenous PE is mediated by activity in the ventral striatum. Specifically, we conducted a parametric modulation analysis to examine whether trial-by-trial variations in striatal activity can be accounted for by the model-derived time series of the PE. Given that the simulations of the episodes unfolded over the 8s of that task period, we modeled the activity with a boxcar regressor that covers the whole of this duration. This analysis was significant in our *a priori* region of interest (ROI), an anatomical mask of the ventral striatum (mask from Oxford-GSK-Imanova Structural-anatomical Striatal Atlas³⁸) ($t_{48} = 5.66$, $p < 0.001$, $d = 0.81$; Fig. 3a). Moreover, the activity in this area showed a poorer fit with an alternative model that codes for the simulation period as a transient event. Considering that this alternative model should be more sensitive to activity evoked by the external presentation of the scenario, the model comparison further suggests that the activity reflects the unfolding internal simulation (Table S2).

Methods, p. 20

Vividness Classifier. We unobtrusively assessed the vividness of the individual episodic simulations by using a neural signature of the vividness of prospective thoughts³⁷. This signature is a whole-brain least-squares regression weight map that provides an estimate of the experienced vividness based on the observed brain activity on a given trial. We first registered the data to the signature map and then calculated a separate GLM for each trial^{72,73}. Each model comprised a regressor coding for a respective trial, and four regressors coding for all other trials as a function of the condition (LR vs HR) and scenario type (pleasant vs neutral-to-unpleasant). As before, the regressors coded for the 8 s of the simulation. This was done separately for each run. We then smoothed the resulting parameter maps with a Gaussian kernel of 8 mm FWHM as in³⁷, and calculated the dot product of the parameter maps and the signature map. The data were then aggregated by condition and participant and submitted to a

paired-samples t-test. We also ran a linear mixed-effects model with decoded vividness and condition (HR vs. LR) as fixed effects and the participant as a random effect to predict absolute pleasantness across individual trials. We followed-up on the significant interaction in this analysis with two separate models for the HR and LR conditions (see also Fig. S6).

Supplement, p. 10

6. Transient or sustained modulation of striatal activity?

We performed Bayesian Model Comparison (BMC) of two model variants. Model 1 (“boxcar model”) is the model underlying our ventral striatum results in Fig 3a and described in the Methods section. Model 2 (“impulse model”) is identical to model 1, with the exemption that the simulation period and, by extension, its parametric modulation by PE is modelled with an event duration of 0s, rather than the entire length of the simulation period (8s).

First, we computed whole-brain BIC maps for each participant and model using the MACS toolbox (Soch & Allefeld, 2018). Next, we computed average BIC values for each participant and model across the mask of the ventral striatum. Finally, we performed BMC on the subject- and model-specific BIC values via the VBA toolbox (Daunizeau et al., 2014) (Table S2). This analysis indicates that, on a group level as well as in the majority of individual participants, the boxcar model outperforms the impulse model. We thus conclude that the ventral-striatum response in Fig 3a more likely reflects a sustained effect as a consequence of the evaluation of the unfolding mental simulation.

Table S2: fMRI model comparison

Model	BIC	Number favoring	Exceedance probability	Model frequency
boxcar	3289.57 ± 42.5	33/49	>0.999	0.736
	161189			
impulse	3291.58 ± 42.2	16/49	<0.001	0.264
	161287			

Note. Shown for each model: Bayesian Information Criteria (BIC) averaged across each participant’s ventral-striatum mask, denoted are mean ± standard error of the mean as well as sum over participants; the number of subjects favoring each model based on BIC scores; exceedance probability and model frequency from the Bayesian Model Comparison.

Comment 2:

Line 243, as far as I understood the experimental design correctly, the same person was always associated with LR or HR (hence 'stable' two-armed bandit). How then is it possible to calculate representations of different people within LR and HR conditions separately?

Our reply: Thank you for pointing out this ambiguity. The reviewer is right to assume that a given person was consistently associated with either the LR or HR conditions. Importantly, as previously described in the methods (p. 15), we had not just one but two persons allocated to each condition. This made it possible to conduct the analysis separately within the two conditions.

Action taken: In the revised results (p. 10), we have changed the wording to emphasize that there were two persons in each condition. We have also updated the corresponding section in the Methods (p. 19-20).

Results, p. 10

Specifically, we employed a region-of-interest that was defined by the *Neurosynth* meta-analysis for the term *people*. In this ROI, we examined whether the neural pattern-similarity between episodes featuring the same person (same-person similarity) is greater than the neural pattern-similarity between episodes featuring different persons (different-person similarity). To avoid any influence of the experimental condition on the results, the different-person similarity only considered the similarity of the respective two persons within a given reward condition. Consistent with our hypothesis, this effect was significant in the dmPFC ($t_{48} = 3.47$, $p < 0.001$, $d = 0.50$; Fig. 3b, left panel).

Methods, p. 19-20

We then quantified the degree to which neural pattern similarity in the dmPFC was consistent with our prediction, namely that it codes for representations of individual people (Figure 3B, left panel, top). In a first step, we created a 16x16 model representational similarity matrix (model RSM) containing predicted pairwise similarities between the 16 neural patterns (4 persons x 4 fMRI runs). Same-person, different-run pairs were coded as 1; different-person, different-run pairs were coded as -1. Because same-person pairs are always from the same reward condition, we likewise restricted the different-person pairs to pairs from the same reward condition. This was done to ensure that any difference with the same-person similarity is not a result of general condition differences (in particular those related to the value of the imagined event).

In the next step, we computed the similarity in activity patterns between all pairs of persons and runs across all voxels of the dmPFC ROI, using Kendall τ^71 . This likewise resulted in a 16x16

neural RSM. Lastly, for each participant, we computed Kendall τ correlations between the model and neural RSMs. Correlation coefficients greater than zero thus indicate that this ROI represents episodes featuring the same person as more similar than episodes featuring different persons. We complemented this anatomical RSA analysis with a spatially unconstrained whole-brain searchlight RSA (see Supplement Tables S7 and S8, and Fig. S9).

Minor comments:

Comment 3:

On line 147, the reported analysis seems to be a convoluted way of performing a repeated measures ANOVA with time and reward condition as within-subject variables. Why is this alternative approach used? Furthermore, if models are being updated based on reward, why wouldn't this also be reflected in a decrease in liking for the LR people?

Our reply regarding the **first part** of the question: We have opted for this analysis approach to clearly highlight the three important comparisons: The pre to post change in liking for the two conditions as well as the difference in this effect between the two conditions. We further examined this effect in relation to the baseline condition to control for any changes that may simply occur as a passage of time. We thus think that this approach constitutes the most parsimonious and targeted test of our hypothesis. Note that we used Wilcoxon tests whenever a Shapiro-Wilk test indicated a deviation from normality.

However, we have also computed a repeated-measures ANOVA with the factors *condition* (HR, LR) and *time* (pre, post). This analysis yielded the identical result pattern. That is, the interaction of condition and time was significant ($F(1,48)=8.18$, $p=.006$, $\eta_p^2=.146$), and follow-up paired t-tests confirmed a positive shift in liking from the pre to post test for the HR ($t_{48}=2.66$, $p=.011$, $d_z=0.38$) but not LR condition ($t_{48}=0.53$, $p=.601$, $d_z=0.08$). Moreover, there was no significant difference between HR and LR on the pre ($t_{48}=0.39$, $p=.696$, $d_z=0.06$), but only on the post-test ($t_{48}=2.68$, $p=.001$, $d_z=0.38$). Given the redundancy with the reported analysis, we have opted to not include these analyses in the revised manuscript. Of course, we'd be happy to add it, if requested by the reviewer and editor.

Our reply regarding the **second part** of the question: We agree that the described mechanism may also lead to a decrease in liking for the people in the LR condition. However, there are several possible reasons for the absence of evidence for such an effect. First, as described in response to comment 1, the neural data indicated that episodes were simulated less vividly in the LR than in the HR condition. Given that the vivid experience of the prospective event is thought to be the driver of learning, we would expect less learning in the former than in the latter condition. Second, whereas the majority of the scenarios in the HR condition were pleasant, scenarios in the LR condition ranged more widely from unpleasant to neutral. We suggest that a condition with a clear selection of more negative scenarios may also yield a more negative shift in liking. Third, in Paulus et al., (2022), we have shown that simulation-based learning can be based on two complementary mechanisms: the more specific one that relies on the valence of the imagined episode and a generic one that, akin to mere exposure, renders the contents of repeatedly imagined events more positive, irrespective of the valence of the

episode. We suggest that this latter effect may have offset any possible downward shift in liking in the LR condition. Indeed, this was our expectation going into the experiment.

Action taken: We have discussed this topic in the revised manuscript (p. 13).

Discussion, p. 13

Another avenue will be to further probe whether the described mechanism of simulation-based learning can also lead to a devaluation, i.e., a decrease in liking. In the current study, persons in the LR condition had not become more disliked at the end of the session, although they were experienced in a number of unpleasant-to-neutral episodes. This absence of evidence may suggest that simulation-based learning can only lead to an upward shift in value. However, we think that it may rather be accounted for by other factors. First, simulations in the LR condition were less vivid and thus likely induced a lower experienced reward as a driver of learning. Another possibility is the valence of the provided scenarios. Whereas the majority of the scenarios in the HR condition were positive, they ranged more widely, from unpleasant to neutral, in the LR condition (Fig. S1). A selection of more negative scenarios might also yield a more negative shift in liking. Third, we have previously shown that simulation-based learning can be based on two complementary mechanisms¹⁸: a more specific one that relies on the valence of the imagined episode and a generic one that, akin to mere exposure, renders the contents of repeatedly imagined events more positive - irrespective of the valence of the episode. We suggest that this latter effect may have offset any of the former effect in the LR condition.

Comment 4:

Missing p-values for Spearman correlations on line 162.

Our reply: The robust correlation toolbox (Pernet et al., 2013) used to compute the skipped Spearman correlation does not provide p-values. Instead, it bases statistical inferences on the bootstrapped 95% confidence intervals, and whether or not it includes zero. This was not the case in the present study: $r_s = 0.38$, 95% CI = [0.11 0.61]. However, for comparison, we also conducted traditional Spearman correlations. The results were as follows: HR–LR update vs. p(HR): $r_s = .45$, $p = .001$; HR update vs. p(HR): $r_s = .41$, $p = .004$.

Reviewer #2 (Remarks to the Author):

Dabas et al. adapted the logic of two-armed bandit tasks to a “social imagination” task. On each trial, participants were asked to choose between persons that they knew and had previously rated as rather neutral. Participants were then asked to imagine interacting with the chosen persons in more or less

pleasant scenarios. Crucially, the probability of a person being linked to pleasant scenarios differed such that they were designed to entail high or low reward. For these imagination phases, participants learned to choose “high reward persons” more often than “low reward persons,” which was reflected in overall changes of linking ratings (pre vs. post the learning task) and in a standard simple Rescorla Wagner model. The prediction errors from this winning model were related to fMRI signals in anatomically defined ROIs of the striatum. As shown by RSA, the to-be-imagined persons were related to fMRI signals in a part of the dmPFC (defined via neurosynth). Signals in this region were related to the Q value of the persons derived from the Rescorla Wagner model. PPI showed a coupling between the two regions.

In our view, this is a very clever and elegant study that shows how imagined social interactions with other persons can drive the evaluation of these persons. Analyses seem to be well-conducted and rather comprehensive given the study design. We could not detect any obvious flaws in the methodological approach, which is state-of-the-art in this field. The paper is very clearly written.

Here, a couple of questions and comments:

Comment 1:

1. Involved brain regions:

a. We were surprised that the authors did not mention parts of the MTL when motivating their design in the introduction. They do find hippocampal activity in whole brain analyses but do not show maps and only very briefly mention this in the discussion. Could you please expand this a bit more or explicitly mention why you do not focus on this region?

Our reply: In this study we focused on the ventral striatum as the canonical region associated with reward learning. By this, we focused on a region that is not strongly associated with episodic simulation *per se* - in contrast to the hippocampus. The striatal results thus provide clear evidence that the reward-learning system is also recruited in simulation-based learning. However, as already discussed in the previous version of the manuscript, we appreciate that the hippocampus is also consistently involved in learning from prediction errors. We were happy to extend the associated discussion in the revised manuscript (p. 12).

Action taken: Specifically, based on the Lisman & Grace (2005) model, we elaborate how the hippocampus plays a critical role in the detection of novelty, which then may be signaled to the ventral striatum. Importantly, we suggest that, in case of simulation-based learning, the hippocampus supports a dual role: It may be engaged first for the construction of the specific episode and then for its evaluation in terms of relative novelty. This may require a big-loop recurrence, where the output of the construction process re-enters the system for subsequent evaluation.

Discussion, p. 12

However, we do not suggest that it is just the striatum that mediates endogenous reinforcement learning. Our exploratory whole-brain analysis identified a network of regions, including the

ventral striatum, anterior hippocampus, and ventromedial prefrontal cortex. Activity in all of these regions is typically associated with the magnitude of a prediction error in neuroimaging studies. In particular, the hippocampus has been suggested to play a critical role by matching incoming information with internal representations. This process is suited for detecting novelty, which is then signaled to midbrain dopamine neurons via the ventral striatum⁴⁵. Striatal^{30,31} and, in particular, dopaminergic activity^{24,46} then induce plasticity in cortical representations and thus afford learning. Notably, the hippocampus may support a dual-role in the case of simulation-based learning. Given its critical involvement in the construction of a coherent scene^{11,12}, it may first foster the simulation of the very episode that is then being evaluated for relative novelty. This process may be mediated via big-loop recurrence⁴⁷, where the output of the simulation is relayed to the cortex before it then reenters the hippocampus for evaluation.

b. Overall brain maps should be shown for the whole brain analyses and discussed in a bit more detail. Tables are more difficult to parse.

Our reply: Please note that we had already made all second level *t*-maps available on Neurovault (<https://identifiers.org/neurovault.collection:17353>) with the previous version of the manuscript. This allows anyone to examine the results in greatest detail and without any arbitrary threshold. However, to make the manuscript more self-contained, we are of course happy to provide additional brain maps in the supplement.

Action taken: In the revised supplement, we complement the tables with further slices to illustrate the whole-brain results (Fig. S7, S8, S9, S10). These also highlight the consistency with our ROI analyses. As mentioned above, we have moreover extended the discussion of the possible hippocampal contribution. However, we are hesitant to further extent the discussion of other areas, given the pitfalls of reverse inferences (Poldrack, 2008).

c. In the figure headings, it is not quite clear that the shown regions are ROIs (at least for the dmPFC).

Our reply and action taken: Thanks for pointing out this ambiguity. We have now added ROI to the respective plots in Fig. 3 and Fig. 4.

Comment 2:

2. RSA:

a. The authors write: “To avoid any influence of the experimental condition on the results, we calculated the different-person similarity separately for the HR and LR conditions.” This is nice but we

did not understand how the authors then calculated the overall pattern. Were the two conditions merged? Did they differ?

Our reply: We apologize that our description of the analysis was not sufficiently detailed. Essentially, we built a model representational-similarity matrix (RSM) that was weighted to reflect our hypothesis. We then correlated this RSM with the neural RSM derived from the fMRI data in the dmPFC.

Action taken: For the revised manuscript, we have now extensively rewritten and extended the description of our approach in the methods (p. 19-20), the results (p. 10), and in the legend to Fig. 3.

Methods, p. 19-20

We then quantified the degree to which neural pattern similarity in the dmPFC was consistent with our prediction, namely that it codes for representations of individual people (Figure 3B, left panel, top). In a first step, we created a 16x16 model representational similarity matrix (model RSM) containing predicted pairwise similarities between the 16 neural patterns (4 persons x 4 fMRI runs). Same-person, different-run pairs were coded as 1; different-person, different-run pairs were coded as -1. Because same-person pairs are always from the same reward condition, we likewise restricted the different-person pairs to pairs from the same reward condition. This was done to ensure that any difference with the same-person similarity is not a result of general condition differences (in particular those related to the value of the imagined event).

In the next step, we computed the similarity in activity patterns between all pairs of persons and runs across all voxels of the dmPFC ROI, using Kendall τ^{71} . This likewise resulted in a 16x16 neural RSM. Lastly, for each participant, we computed Kendall τ correlations between the model and neural RSMs. Correlation coefficients greater than zero thus indicate that this ROI represents episodes featuring the same person as more similar than episodes featuring different persons. We complemented this anatomical RSA analysis with a spatially unconstrained whole-brain searchlight RSA (see Supplement Tables S7 and S8, and Fig. S9).

Results, p. 10

Specifically, we employed a region-of-interest that was defined by the *Neurosynth* meta-analysis for the term *people*. In this ROI, we examined whether the neural pattern-similarity between episodes featuring the same person (same-person similarity) is greater than the neural pattern-similarity between episodes featuring different persons (different-person similarity). To avoid any influence of the experimental condition on the results, the different-person similarity only considered the similarity of the respective two persons within a given reward condition.

Consistent with our hypothesis, this effect was significant in the dmPFC ($t_{48} = 3.47$, $p < 0.001$, $d = 0.50$; Fig. 3b, left panel).

b. It is not quite clear how many events were used. Were some of them repeated? Wouldn't it be possible to run RSA on the similarity (in pleasantness) of the events? Would you expect this similarity to be captured in the same ROI? ... maybe along with the general pleasantness ratings?

Our reply and action taken: We appreciate the need for further details here, which we have provided in the revised methods. Indeed, towards the end of the session (that was comprised of 96 trials), we had to repeat a limited number of positive events for 29/49 participants (mean = 4.48, range = 1 to 7) and of neutral-to-unpleasant events for 6/49 participants (mean = 1, range = 1 to 1). We have reported these numbers in the revised methods section (p. 15).

Methods, p. 15

The people in the HR condition were imagined in pleasant scenarios with a higher probability than those in the LR condition (80% vs. 30% of the trials). In the remaining trials, participants imagined interacting with the selected person in one of the neutral-to-unpleasant scenarios.

The specific scenario was randomly chosen for any given trial. Towards the end of the 96 trials of a session, we had to repeat a limited number of scenarios in a number of participants (pleasant events in 29 participants: mean number of repetitions = 4.48, range 1 to 7; neutral-to-unpleasant events in 6 participants, mean = 1, range = 1 to 1). In some of those few cases, the same scenario was imagined with two different people.

Re. the queried RSA of the pleasantness, we assume that such an analysis could also identify regions that are typically involved in the representation of reward. As such, they may show some overlap with the highlighted regions, such as the striatum and also parts of the medial prefrontal cortex.

Comment 3:

3. Events: In general, some more details would be nice in the main texts and especially in the SI.

a. It would be important to have a list of the events along with the ratings of these from the two samples.

b. What was the variability in ratings? Some scenarios might be more positive/negative (e.g., depending on whether you like ice cream)

Our reply to comment 3 a and 3b and action taken: We agree with the importance of providing the stimulus material. Accordingly, we have now added a spreadsheet with all events to our OSF project repository (scenarios_ratings.csv). It includes the mean and standard deviation (SD) of the pleasantness rating of the two studies (i.e., pilot and study proper). The scenarios do somewhat differ in the SD of the ratings (SD range for pleasant scenarios: 0.11 to 0.25 and for unpleasant-to-neutral scenarios: 0.05 to 0.31). The revised manuscript links to this file on p. 15.

Methods, p. 15

With an independent sample of participants (n = 107), we validated a set of 129 sentences describing scenarios that are either pleasant (67) or neutral-to-unpleasant (62) and that can be imagined in detail (see Supplement for details). The sentences served to induce the simulations of specific episodes in the MRI scanner. [All scenarios along with their pleasantness ratings can be found on the project's OSF repository at https://osf.io/q3v6b/?view_only=625d2433441d41019707903672c71ce5.](https://osf.io/q3v6b/?view_only=625d2433441d41019707903672c71ce5)

c. Can the events be categorized into more or less mundane, frequent, special events? E.g., eating ice can be imagined with any person but going to the opera or a metal concert might be difficult to imagine with same people. Relatedly, does the difficulty/oddness/vividness of the imagination have an influence on how "rewarding" it is? As far as we see, the authors did not specifically assess this but 1-2 lines of discussion would be good.

Our reply: In general, we had selected scenarios that are fairly easy to imagine with a wide range of people. For example, one scenario asks participants to imagine themselves with the respective person in the VIP section of a concert. However, it does not specify the kind of concert to accommodate a range of preferences.

We also think that the reviewers are spot on in asking the questions about difficulty/oddness/vividness. Indeed, we had similar thoughts regarding avenues for future research. Specifically, as mentioned above, our new analyses indicate that more vivid imaginations tend to be more rewarding. In turn, imaginations are easier to build up and more vivid if they take place in more familiar locations (Robin, 2018; Robin & Moscovitch, 2014). This is presumably because more familiar places provide a better scaffold for the mental construction of the simulated episode. Thus, boosting vividness by providing more familiar locations might further facilitate learning. (Please see our response to next comment for additions to the discussion.)

d. Related to the previous point, are there often causal links between event and person implied (e.g., Sally has broken your bike or she just has found it? Has Harry invited you to eating ice cream or have you just met him coincidentally?). Again, 1-2 lines of discussion could be interesting.

Our reply: Again, we think this is a really interesting question. The majority of our scenarios did not involve a clear causal link between event and person. However, we would hypothesize that learning would not only be stronger but potentially also qualitatively different, if such causalities were implied. This is clearly another avenue for future research, which we were happy to incorporate in the new discussion.

Action taken in response to comments 3 c & d: We have discussed these ideas in the revised manuscript (p. 13).

Discussion, p. 13

An important avenue for future research will be to examine how features of the imagined episode influence the ensuing learning. For example, in the majority of the scenarios, the imagined person did not cause the specific event. Learning may not only be stronger but also qualitatively different, if the scenarios were to imply such causality or intentionality^{53–55}. Moreover, we observed that more vivid simulations (as decoded from brain activity) were associated with stronger affective experiences in the high-reward condition. This suggest that learning could potentially be boosted by enhancing the vividness of the simulation. This could be achieved, for example, by manipulating the familiarity of the locations at which the episodes are being imagined^{56,57}.

e. Did all participants see the very same events? How was the order randomized? Did they see some events twice? ... with different persons? See comment above for RSA.

Our reply: The events were randomly drawn from the same pool for each participant, with the constraint that any event could only be repeated once all others of the same kind (i.e., pleasant or neutral-to-unpleasant) had been presented. In some of those few cases, the same scenario was imagined with two different people. In addition to describing the number of event repetitions (please see response to comment 2b), we now also state that event allocation was randomized.

Action taken: We describe the scenario allocation in more detail in the revised methods (p. 15).

Methods, p. 15

The specific scenario was randomly chosen for any given trial. Towards the end of the 96 trials of a session, we had to repeat a limited number of scenarios in a number of participants (pleasant events in 29 participants: mean number of repetitions = 4.48, range 1 to 7; neutral-to-unpleasant events in 6 participants, mean = 1, range = 1 to 1). In some of those few cases, the same scenario was imagined with two different people.

Comment 4:

4. Persons:

a. Did the imagination also change the rated familiarity with these persons?

Our reply: In response to this query, we now also report simulation-induced changes in familiarity in the supplement. Similar to previous studies (Szpunar & Schacter, 2013; Garcia Jimenez et al., 2023), simulations increased the perceived familiarity of the imagined persons in both the HR ($t_{48} = 2.35$, $p = .01$, $d = .33$) and the LR ($W = 788$, $p = .04$; Shapiro Wilk $W = .92$, $p = .002$) conditions. Notably, this effect did not differ between the groups, indicating that it was based on a different mechanism than

the value update. This study thus adds to those other reports that repeated simulations can increase the perceived plausibility and likelihood of experiencing an event with a particular person (Garcia Jimenez et al., 2023; Szpunar & Schacter, 2013).

Action taken: In the revised supplement, we describe the impact of simulations on familiarity (Section 4).

Supplement, p. 8

4. Episodic simulation induces change in familiarity

We also assessed whether repeated simulations change the perceived familiarity of the imagined persons. Participants therefore rated the familiarity of each HR and LR person before and after the simulation task. They also rated two people who were not part of the task, serving as a baseline condition. To quantify a change in familiarity, we subtracted the pre-task familiarity ratings from the post-task ratings. We then corrected the change scores for the HR and LR conditions by subtracting those of the baseline condition.

We observed an increase in familiarity for both HR ($t_{48} = 2.35, p = .01, d = .33$) and LR ($W = 788, p = .04$; Shapiro Wilk $W = .92, p = .002$) people (Fig. S5). Notably, the difference between the HR and LR conditions was not significant ($t_{48} = 1.36, p = .18, d = .19$), unlike for the liking ratings.

This finding aligns with previous research showing that repeated simulations can enhance the perceived plausibility or likelihood of experiencing an event with a particular person (Garcia Jimenez et al., 2023; Szpunar & Schacter, 2013).

b. One of the crucial points related to the previous point that we did not quite understand is: Since participants learned to choose the HR persons more often than the LR persons, did they also get to imagine more scenarios with the HR persons? So, to what degree are effects driven by the frequency of imagination versus the pleasantness of the imagined scenario? What would happen if participants were “forced” to imagine each person with the same frequency but with more or less pleasant events. Instead of an explicit choice, participants could have given a rating to fit the model. A few lines of discussion would be interesting.

Our reply: This is an interesting question. In principle, both of these effects can potentially contribute to simulation-based learning, and a goal for future studies may be to further disentangle their relative contributions. However, for several reasons, we think that the current data are difficult to account for without assuming that the experienced pleasantness was a driving force for learning. First, we have previously shown that the valence of the simulated episodes determines learning, even when we kept frequency constant across conditions (Benoit et al., 2019; Paulus et al., 2022). Second, in the current study, we have shown that the model that can account best for the learning is based on the experienced pleasantness (via the vividness of the simulation; see response to Reviewer 1’s comment

1) and the ensuing prediction error. Together, these two lines of evidence indicate that the experienced pleasantness underlies the observed simulation-based learning.

Action taken: In the revised discussion, we now refer to the past work and highlight the contribution of the experienced pleasantness, rather than just the frequency, to simulation-based learning.

Discussion, p. 12

The endogenous prediction error seems to drive learning much like experience-based prediction errors. Specifically, we found that this learning can be described as an effort to minimize these errors as formalized by the Rescorla-Wagner model. This simple model was better at accounting for the data than a number of alternative models of various complexity. The current data thus add to previous reports^{18,19} that learning is not just driven by the frequency of simulations, but by the mentally experienced reward during those simulations and the ensuing prediction error. Over the years, the Rescorla-Wagner model has been shown to account for diverse phenomena – including Pavlovian and instrumental learning of simple features of the environment up to learning from complex social interactions^{43,44}. Here, we generalize those findings by showing that there is no need for an actual reward to trigger learning.

Comment 5:

5. Did the authors preregister parts of the study?

Our reply: We did not preregister this study. However, we note that parts of it are conceptual replications of the studies reported in Paulus et al. (2022) as well as Benoit et al. (2019) (one of which was preregistered). We thus suggest that there is a solid empirical basis for the basic effect of simulation-based learning. We have also emphasized this in the revised Results.

Results, p. 6

To examine the change in liking, we subtracted the liking ratings of the initial test from the one following the simulation task. We then corrected the change scores for the HR and LR conditions by subtracting the change scores of the baseline condition. This measure did not yield a significant change in liking for the LR people ($t_{48} = 0.54$, $p = 0.59$, $d = 0.08$). By contrast, and consistent with our hypothesis, it revealed an increase in liking for the HR people ($W = 958$, $p < 0.001$, $r = 0.49$; Shapiro Wilk: $W = 0.94$, $p = 0.02$). This increase was more pronounced than the absent effect for the LR people ($W = 875$, $p = 0.004$, $r = 0.37$; Shapiro Wilk: $W = 0.95$, $p = 0.03$; Fig. 2c). This effect thus conceptually replicates and extends earlier work^{18,19}.

Comment 6:

6. Parameter recovery in Fig. S2. What was the range of tested parameters?

Our reply: The range of the tested parameters were as follows:

Model	Parameter	Lower threshold	Upper threshold
Win-stay lose-shift	Epsilon	0	1
Rescorla-Wagner (RW)	Alpha	0.05	1
	Beta	0	25
Choice Kernel (CK)	Alpha	0.05	1
	Beta	0	25

We used the same parameter ranges for the combined RW-CK model as for the individual RW and CK models.

Action taken: We have added this table to the Supplement (section 2.ii Parameter Recovery) as Table S1.

Comment 7:

7. Figure 2 “simulations:” This word is confusing because it sounds as if the data were simulated. Imagination might be better.

Our reply: We agree and have changed this accordingly to “imagination”.

Reviewer #3 (Remarks to the Author):

Our reply: We also thank the third reviewer for dedicating their time for the evaluation of our manuscript.

End of replies to reviewer comments

In closing, we again would like to thank the reviewers and editor for their encouraging and constructive feedback. We think that our revisions in response to those comments helped considerably in strengthening our manuscript. We hope that you now deem it suitable for publication.

References

- Abdulrahman, H., & Henson, R. N. (2016). Effect of trial-to-trial variability on optimal event-related fMRI design: Implications for Beta-series correlation and multi-voxel pattern analysis. *NeuroImage*, *125*, 756–766.
- Ames, D. L., & Fiske, S. T. (2015). Perceived intent motivates people to magnify observed harms. *Proceedings of the National Academy of Sciences*, *112*(12), 3599–3605.
<https://doi.org/10.1073/pnas.1501592112>
- Bao, S., Chan, V. T., & Merzenich, M. M. (2001). Cortical remodelling induced by activity of ventral tegmental dopamine neurons. *Nature*, *412*(6842), Article 6842.
<https://doi.org/10.1038/35083586>
- Behrens, T. E. J., Hunt, L. T., & Rushworth, M. F. S. (2009). The Computation of Social Behavior. *Science*, *324*(5931), 1160–1164. <https://doi.org/10.1126/science.1169694>
- Benoit, R. G., Paulus, P. C., & Schacter, D. L. (2019). Forming attitudes via neural activity supporting affective episodic simulations. *Nature Communications*, *10*(1), Article 1.
<https://doi.org/10.1038/s41467-019-09961-w>
- Daunizeau, J., Adam, V., & Rigoux, L. (2014). VBA: A probabilistic treatment of nonlinear models for neurobiological and behavioural data. *PLoS Computational Biology*, *10*(1), e1003441.
<https://doi.org/10.1371/journal.pcbi.1003441>
- Den Ouden, H. E. M., Daunizeau, J., Roiser, J., Friston, K. J., & Stephan, K. E. (2010). Striatal Prediction Error Modulates Cortical Coupling. *The Journal of Neuroscience*, *30*(9), 3210–3219. <https://doi.org/10.1523/JNEUROSCI.4458-09.2010>
- Garcia Jimenez, C., Mazzoni, G., & D'Argembeau, A. (2023). Repeated simulation increases belief in the future occurrence of uncertain events. *Memory & Cognition*, *51*(7), 1593–1606.
<https://doi.org/10.3758/s13421-023-01414-6>
- Garvert, M. M., Moutoussis, M., Kurth-Nelson, Z., Behrens, T. E. J., & Dolan, R. J. (2015). Learning-Induced Plasticity in Medial Prefrontal Cortex Predicts Preference Malleability. *Neuron*, *85*(2), 418–428. <https://doi.org/10.1016/j.neuron.2014.12.033>

- Hassabis, D., Kumaran, D., Vann, S. D., & Maguire, E. A. (2007). Patients with hippocampal amnesia cannot imagine new experiences. *Proceedings of the National Academy of Sciences*, *104*(5), 1726–1731. <https://doi.org/10.1073/pnas.0610561104>
- Koster, R., Chadwick, M. J., Chen, Y., Berron, D., Banino, A., Düzel, E., Hassabis, D., & Kumaran, D. (2018). Big-Loop Recurrence within the Hippocampal System Supports Integration of Information across Episodes. *Neuron*, *99*(6), 1342-1354.e6. <https://doi.org/10.1016/j.neuron.2018.08.009>
- Lee, S., Parthasarathi, T., Cooper, N., Zauberman, G., Lerman, C., & Kable, J. W. (2022). A neural signature of the vividness of prospective thought is modulated by temporal proximity during intertemporal decision making. *Proceedings of the National Academy of Sciences*, *119*(44), e2214072119. <https://doi.org/10.1073/pnas.2214072119>
- Lee, S., Parthasarathi, T., & Kable, J. W. (2021). The Ventral and Dorsal Default Mode Networks Are Dissociably Modulated by the Vividness and Valence of Imagined Events. *The Journal of Neuroscience: The Official Journal of the Society for Neuroscience*, *41*(24), 5243–5250. <https://doi.org/10.1523/JNEUROSCI.1273-20.2021>
- Lisman, J. E., & Grace, A. A. (2005). The hippocampal-VTA loop: Controlling the entry of information into long-term memory. *Neuron*, *46*(5), 703–713. <https://doi.org/10.1016/j.neuron.2005.05.002>
- Mumford, J. A., Davis, T., & Poldrack, R. A. (2014). The impact of study design on pattern estimation for single-trial multivariate pattern analysis. *Neuroimage*, *103*, 130–138.
- Nili, H., Wingfield, C., Walther, A., Su, L., Marslen-Wilson, W., & Kriegeskorte, N. (2014). A Toolbox for Representational Similarity Analysis. *PLOS Computational Biology*, *10*(4), e1003553. <https://doi.org/10.1371/journal.pcbi.1003553>
- Paulus, P. C., Dabas, A., Felber, A., & Benoit, R. G. (2022). Simulation-based learning influences real-life attitudes. *Cognition*, *227*, 105202. <https://doi.org/10.1016/j.cognition.2022.105202>

- Pernet, C., Wilcox, R., & Rousselet, G. (2013). Robust Correlation Analyses: False Positive and Power Validation Using a New Open Source Matlab Toolbox. *Frontiers in Psychology, 3*.
<https://www.frontiersin.org/articles/10.3389/fpsyg.2012.00606>
- Poldrack, R. A. (2008). The role of fMRI in cognitive neuroscience: Where do we stand? *Current Opinion in Neurobiology, 18*(2), 223–227. <https://doi.org/10.1016/j.conb.2008.07.006>
- Race, E., Keane, M. M., & Verfaellie, M. (2011). Medial Temporal Lobe Damage Causes Deficits in Episodic Memory and Episodic Future Thinking Not Attributable to Deficits in Narrative Construction. *Journal of Neuroscience, 31*(28), 10262–10269.
<https://doi.org/10.1523/JNEUROSCI.1145-11.2011>
- Reynolds, J. N. J., & Wickens, J. R. (2002). Dopamine-dependent plasticity of corticostriatal synapses. *Neural Networks, 15*(4–6), 507–521. [https://doi.org/10.1016/S0893-6080\(02\)00045-X](https://doi.org/10.1016/S0893-6080(02)00045-X)
- Robin, J. (2018). Spatial scaffold effects in event memory and imagination. *WIREs Cognitive Science, 9*(4), e1462. <https://doi.org/10.1002/wcs.1462>
- Robin, J., & Moscovitch, M. (2014). The effects of spatial contextual familiarity on remembered scenes, episodic memories, and imagined future events. *Journal of Experimental Psychology: Learning, Memory, and Cognition, 40*(2), 459–475.
<https://doi.org/10.1037/a0034886>
- Soch, J., & Allefeld, C. (2018). MACS - a new SPM toolbox for model assessment, comparison and selection. *Journal of Neuroscience Methods, 306*, 19–31.
<https://doi.org/10.1016/j.jneumeth.2018.05.017>
- Szpunar, K. K., & Schacter, D. L. (2013). Get real: Effects of repeated simulation and emotion on the perceived plausibility of future experiences. *Journal of Experimental Psychology: General, 142*(2), 323–327. <https://doi.org/10.1037/a0028877>
- Thorwart, A., & Livesey, E. J. (2016). Three Ways That Non-associative Knowledge May Affect Associative Learning Processes. *Frontiers in Psychology, 7*.
<https://doi.org/10.3389/fpsyg.2016.02024>

- Tziortzi, A. C., Searle, G. E., Tzimopoulou, S., Salinas, C., Beaver, J. D., Jenkinson, M., Laruelle, M., Rabiner, E. A., & Gunn, R. N. (2011). Imaging dopamine receptors in humans with [11C]-(+)-PHNO: Dissection of D3 signal and anatomy. *NeuroImage*, *54*(1), 264–277.
<https://doi.org/10.1016/j.neuroimage.2010.06.044>
- Undeger, I., Visser, R. M., & Olsson, A. (2020). Neural Pattern Similarity Unveils the Integration of Social Information and Aversive Learning. *Cerebral Cortex*, *30*(10), 5410–5419.
<https://doi.org/10.1093/cercor/bhaa122>
- Zhang, L., Lengersdorff, L., Mikus, N., Gläscher, J., & Lamm, C. (2020). Using reinforcement learning models in social neuroscience: Frameworks, pitfalls and suggestions of best practices. *Social Cognitive and Affective Neuroscience*, *15*(6), 695–707.
<https://doi.org/10.1093/scan/nsaa089>

We thank the editor and the three reviewers for their positive feedback. As detailed below, in the revised manuscript, we have carefully addressed the remaining query of Reviewer 1. That is, we have clarified how we conducted the analysis with the vividness classifier.

Start of reply to reviewer comments

Reviewer #1 (Remarks to the Author):

I thank the authors for revising their manuscript in response to my previous comments.

In response to my previous concern that the activations might reflect a response to merely reading the presented text rather than vividly imagining the suggested scenarios, the authors have now added two extra control analyses. The first analysis looks at the dynamics of the neural activations, based on the assumption that if the activation reflects stimulus processing instead of imagination, it would be short lived. The second analysis quantified the experienced vividness of imagination using a ‘vividness-classifier’ on the whole-brain activity and showed that decoded vividness was higher in the high-reward compared to the low-reward condition. These are both excellent control analyses.

However, it was not clear to me how the vividness classifier was defined, given that no vividness ratings were collected in this data set. Did the authors use the classifier that was developed by Lee et al. (2022)? Looking at that paper, I did not find a link to the whole-brain regression weights, but I might have missed it. More explanation on how this analysis was executed is needed to evaluate the validity.

Furthermore, I appreciate that the authors demonstrate that similar statistical conclusions can be obtained with a more standard repeated-measures ANOVA. I do not have a strong preference for including one analysis over the other, my only thought is that the ANOVA might be more straightforward for a lot of readers to interpret.

Besides these, all my other comments have been addressed, and I would like to congratulate the authors on an excellent piece of work.

Our reply: We thank the reviewer for their kind remarks about our work and for highlighting the need for clarification concerning the vividness classifier. The original paper by Lee et al. (2022) does not include a link to the vividness classifier. Upon personal request, we had received the whole-brain regression weights, along with the fMRIprep standard image used for registration, directly from the first author, Dr. Sangil Lee. We note that the authors have now also made the classifier openly available at <https://www.thedecodinglab.com/resources>.

Action taken: We have updated the methods section (p. 20) to clarify this point and to correct minor inaccuracies in the description of the analysis.

Methods from p. 20:

Vividness Classifier. We unobtrusively assessed the vividness of the individual episodic simulations by using a neural signature of the vividness of prospective thoughts³⁷. This

signature is a whole-brain least-squares regression weight map that provides an estimate of the experienced vividness based on the observed brain activity on a given trial. We obtained the classifier upon personal request from the first author. It is also openly available at <https://www.thedecodinglab.com/resources>.

We first resampled and registered the preprocessed data to the standard fMRIprep MNI 2-mm template and, thus, to the original signature map. We then calculated a separate GLM for each trial^{72,73}. Each model comprised one regressor coding for a respective trial and another regressor coding for all other trials. As in the other analyses, the regressors covered the 8 s of the simulation, and motion parameters were included as regressors of no interest. We concatenated the data across the four functional runs, and accordingly adjusted the high-pass filter and temporal non-sphericity estimates as implemented in SPM12.

We then smoothed the resulting parameter maps with a Gaussian kernel of 8 mm FWHM as in³⁷, and calculated the dot product of the parameter maps and the signature map. These dot product values were aggregated by condition (HR vs. LR) and participant, and submitted to a paired-samples t-test. We also ran a linear mixed-effects model with decoded vividness and condition (HR vs. LR) as fixed effects and the participant as a random effect to predict absolute pleasantness across individual trials. We followed up on the significant interaction in this analysis with two separate models for the HR and LR conditions (see also Fig. S6).

Reviewer #1 (Remarks on code availability):

The code is well-documented and clear.

Reviewer #2 (Remarks to the Author):

Thank you very much for addressing my and my co-reviewer's comments. We are confident that this paper will be read widely and will inspire many new studies.

Reviewer #3 (Remarks to the Author):

Our reply to Reviewers #2 and 3: We thank the reviewers for their thoughtful feedback and encouraging words. We much appreciate their time and effort dedicated to help improve our manuscript.

References

Lee, S., Parthasarathi, T., Cooper, N., Zauberman, G., Lerman, C., & Kable, J. W. (2022). A neural signature of the vividness of prospective thought is modulated by temporal proximity during intertemporal decision making. *Proceedings of the National Academy of Sciences*, *119*(44), e2214072119. <https://doi.org/10.1073/pnas.2214072119>

Mumford, J. A., Davis, T., & Poldrack, R. A. (2014). The impact of study design on pattern estimation for single-trial multivariate pattern analysis. *Neuroimage*, *103*, 130–138.

Abdulrahman, H., & Henson, R. N. (2016). Effect of trial-to-trial variability on optimal event-related fMRI design: Implications for Beta-series correlation and multi-voxel pattern analysis. *NeuroImage*, *125*, 756–766.